# INT: SELF-PROPOSED INTERVENTIONS ENABLE CREDIT ASSIGNMENT IN LLM REASONING

**Matthew Y. R. Yang[1], Hao Bai[2], Ian Wu[1], Gene Yang[1], Amrith Setlur[1], Aviral Kumar[1]**
[1]Carnegie Mellon University, [2]University of Illinois Urbana-Champaign

## ABSTRACT

Outcome-reward reinforcement learning (RL) has proven effective at improving the reasoning capabilities of large language models (LLMs). However, standard RL assigns credit only at the level of the final answer, penalizing entire reasoning traces when the outcome is incorrect and uniformly reinforcing all steps when it is correct. As a result, correct intermediate steps may be discouraged in failed traces, while spurious steps may be reinforced in successful ones. We refer to this failure mode as the problem of *credit assignment*. While a natural remedy is to train a process reward model, accurately optimizing such models to identify corrective reasoning steps remains challenging. We introduce **Intervention Training** (`InT`), a training paradigm in which the model performs fine-grained credit assignment on its own reasoning traces by proposing short, targeted corrections that steer trajectories toward higher reward. Using reference solutions commonly available in mathematical reasoning datasets and exploiting the fact that verifying a model-generated solution is easier than generating a correct one from scratch, the model identifies the first error in its reasoning and proposes a single-step *intervention* to redirect the trajectory toward the correct solution. We then apply supervised fine-tuning (SFT) to the on-policy rollout up to the point of error concatenated with the intervention, localizing error to the specific step that caused failure. We show that the resulting model serves as a far better initialization for RL training. After running `InT` and subsequent fine-tuning with RL, we improve accuracy by nearly 14% over a 4B-parameter base model on IMO-AnswerBench, outperforming larger open-source models such as `gpt-oss-20b`.

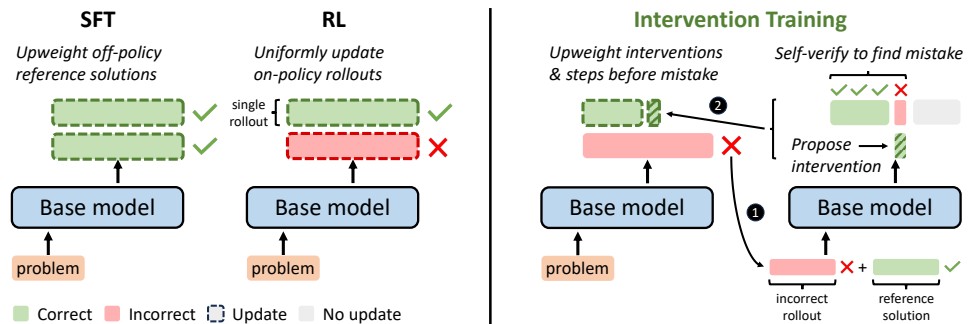

Figure 1: *Intervention training (`InT`) for improving credit assignment.* `InT` proposes *single-step* interventions to replace incorrect intermediate steps in reasoning traces (1). Conditioned on these localized corrections, the model can generate counterfactual continuations that succeed where the original failed (2). We then perform supervised fine-tuning on these interventions, enabling effective credit assignment by upweighting the likelihood of the interventions in place of the mistakes.

## 1 INTRODUCTION

Post-training large language models (LLMs) with reinforcement learning (RL) has proven to be a highly effective strategy for improving their reasoning capabilities. As RL is scaled to more challenging tasks, training becomes increasingly dominated by incorrect rollouts, which are often

long-horizon trajectories composed of many smaller, structured reasoning steps. Many such rollouts contain substantial portions of correct reasoning, yet outcome-based RL treats these correct steps the same as the critical mistakes that cause failure. This indiscriminate penalty can lead to undesirable behaviors including increased verbosity or frequent, premature shifts in the reasoning process, which have been widely observed when applying outcome-reward RL (Chen et al., 2025; Wang et al., 2025b). Moreover, on sufficiently difficult problems where no correct rollout is produced at all, outcome-reward RL provides no learning signals, as advantages collapse to zero.

In principle, addressing this challenge requires solving the problem of *credit assignment*: if we could identify the step at which a reasoning trace goes astray and penalize that particular step while reinforcing other possible promising steps, we could drive the model toward more favorable outcomes. However, existing approaches to credit assignment are difficult to apply in the LLM setting. For reasoning traces, estimating the "credit" (i.e., value) of each step typically requires running multiple branched rollouts conditioned on a given prefix, which is prohibitively expensive (Kazemnejad et al., 2024; Luo et al., 2024; Kim et al., 2025). Prior work attempts to amortize this process by training explicit value functions (or process reward models) (Setlur et al., 2024b; Luo et al., 2024), but training such functions over long reasoning traces remains an open question. Moreover, even with a reasonably accurate value function, optimizing it to identify an alternate step is challenging (Wang et al., 2025a; Zhang et al., 2025c) due to the vast space of possible future steps.

***How can we perform credit assignment without training a value function?*** Rather than explicitly learning a value function, we ask the model itself to identify how failed trajectories can be locally corrected. This approach relies on an LLM's ability to *implicitly* perform value estimation and optimization jointly, in order to propose an alternative, improved step, which we refer to as a ***corrective intervention***. Our idea is to reduce the task of generating end-to-end solutions to the substantially easier task of verifying individual steps, which can be accomplished via textual comparison against a reference solution (a textual "diff"). In our method, we instruct a base model to analyze the differences between a reference solution and an incorrect reasoning trace that it previously generated, with the goal of identifying the first point at which the trace goes wrong and proposing a single-step corrective intervention. When conditioned on this intervention, the model can then generate counterfactual reasoning traces that succeed from right before where the original rollout had failed.

We then train the model to internalize these interventions by applying supervised fine-tuning (SFT) on them, followed by RL. By fine-tuning the model on targeted single-step interventions, we selectively reduce the likelihood of the incorrect reasoning steps that the base model would otherwise produce, shifting it toward more favorable steps instead. Our approach remains notably simple and computationally efficient, as it avoids branched rollouts, explicit value-function training, or modifying the RL objective to include step-level rewards. Throughout this procedure, we never rely on a larger model; instead, we leverage the *asymmetry (Setlur et al., 2025b) in task difficulty* between instruction-following, verification, and generation within the same model to perform credit assignment. We call this approach **Intervention Training** (`InT`).

We compare `InT` to several alternative approaches that leverage reference solutions for learning, including methods that train policies directly on reference solutions. We find that `InT` produces more "on-policy" trajectories (those with high likelihood under the base model), whereas other approaches tend to produce more "off-policy" ones. After online RL, `InT` improves performance by nearly 10% on average across four challenging mathematical reasoning benchmarks, including a $\sim$14% gain on IMO-AnswerBench (Luong et al., 2025), which consists of IMO-level problems curated by former medalists. These results demonstrate **InT** as a simple and effective paradigm for improving the reasoning capabilities of LLMs, through improved credit assignment.

## 2 PRELIMINARIES, NOTATION, AND PROBLEM STATEMENT

**Learning objective.** Training LLMs for reasoning typically involves an LLM $\pi$, model-generated reasoning traces $\mathbf{y} \sim \pi(\cdot|\mathbf{x})$ produced on input prompts $\mathbf{x} \sim \rho$, and a binary reward $r(\mathbf{x}, \mathbf{y}) \in \{0, 1\}$ quantifying the correctness of the final answer. The goal is to maximize average reward, given by:

$$\max_{\pi} \quad J(\pi) := \mathbb{E}_{\mathbf{x} \sim \rho, \mathbf{y} \sim \pi(\cdot|\mathbf{x})}[r(\mathbf{x}, \mathbf{y})].$$

We can maximize the objective by updating the policy with the policy gradient (Williams, 1992):

$$\pi' \leftarrow \pi + \alpha \cdot \mathbb{E}_{\mathbf{x} \sim D_{\text{train}}, \mathbf{y} \sim \tilde{\pi}(\cdot|\mathbf{x})}[r(\mathbf{x}, \mathbf{y}) \cdot \nabla_{\pi} \log \pi(\mathbf{y}|\mathbf{x})], \tag{1}$$

where $D_{\text{train}}$ is the set of training problems and $\tilde{\pi}$ is the policy used to generate samples for training. In policy gradient methods such as GRPO (Shao et al., 2024), $\tilde{\pi} = \pi_{\text{old}}$ is a periodically updated copy of $\pi$, with additional correction terms to account for the distribution shift between $\pi_{\text{old}}$ and $\pi$. We can also express SFT with the same rule by , where $\tilde{\pi}$ represents a pre-collected offline dataset consisting of only correct trajectories, so that Equation 1 effectively maximizes likelihood on these traces.

**Credit assignment.** Reasoning trajectories $\mathbf{y}$ can naturally be decomposed into a sequence of individually complete segments[1], $\mathbf{y} = (\mathbf{y}_0, \mathbf{y}_1, \ldots, \mathbf{y}_T)$, where each reasoning step $\mathbf{y}_t$ contributes unequally to the final answer. Ideally, learning signals should reflect this uneven contribution. However, in Equation 1, the policy is updated with binary rewards that entirely depend on the correctness of the final answer ("outcome rewards"). Outcome-reward RL normalizes these rewards across responses sampled for the same prompt $\mathbf{x}$, yielding *advantages*. When the advantage $A(\mathbf{x}, \mathbf{y}_i) = r(\mathbf{x}, \mathbf{y}_i) - 1/n \sum_{j=1}^{n} r(\mathbf{x}, \mathbf{y}_j)$ is positive, every step in the response $\mathbf{y}_i$ is uniformly reinforced; when it is negative, all steps are uniformly discouraged. With this form of reward allocation, steps that play no role in reaching the solution may still be upweighted, while steps that are locally correct but followed by later mistakes may be suppressed. In long-horizon settings where trajectories $\mathbf{y}_i$ consist of hundreds of steps, the resulting noise can overwhelm the learning signal.

This limitation of outcome-reward RL manifests in two ways. First, when training with large token budgets, rewards often plateau early and do not improve substantially over the course of training (Setlur et al., 2025b; Luo et al., 2025; Yeo et al., 2025). Second, responses become either excessively verbose or prematurely truncated, often failing to maximize the underlying outcome reward (Qu et al., 2025b; Chen et al., 2025; Wang et al., 2025b). Avoiding these issues requires accurately identifying which steps are responsible for success or failure and disentangling correct steps from incorrect responses and incorrect steps from otherwise correct responses.

This problem of determining how individual intermediate steps contribute to the final outcome is known as *credit assignment*. If we can estimate the "credit" of each intermediate step with a step-level value function (also called Process Reward Models, or PRMs), we can update $\pi$ by encouraging the correct parts of a solution while discouraging the incorrect ones. However, this approach faces two major obstacles in practice. First, reliably estimating such value functions is difficult due to its cost (Luo et al., 2024; Setlur et al., 2024b; Kazemnejad et al., 2024) and training instabilities (Wang et al., 2025a; Zhang et al., 2025c). Second, even when a sufficiently accurate PRM is available, simply penalizing incorrect parts of a solution does not ensure that they will be replaced with higher-value, correct alternatives. As a result, policy optimization remains a challenging problem, since efficiently searching for suitable replacement steps is a difficult task in itself.

**Problem statement.** Our goal is to develop an effective approach to credit assignment. In particular, we focus on assigning credit within *incorrect* rollouts by pinpointing the incorrect steps that derail the solution. In our setting, we assume access to a reference solution (e.g., a human-written one) for each training problem, as they are commonly available in existing open-source math datasets.

*Why focus on credit assignment for incorrect traces?* As we move toward more challenging RL tasks, incorrect rollouts dominate trajectories observed during training. For example, Figure 11 illustrates that, on a curated set of Olympiad-level Math problems, more than 80% of rollout groups contain no successful trajectories at the start of training. Being able to extract useful learning signals from these failed rollouts would therefore tap into an abundant source of supervision that would otherwise be completely discarded by outcome-reward RL, as advantages within these groups collapse to zero.

**Where within incorrect trajectories is credit assignment needed?** In Figure 2, we visualize the locations of the *first error* that derails incorrect trajectories. We find that these errors can occur at a diverse set of points throughout the trajectory, often far from the initial steps, with more than 60% of the first errors arising after the 50th step. Consequently, effective credit assignment for incorrect rollouts requires a method that can precisely identify errors occurring at many different points along a trajectory.

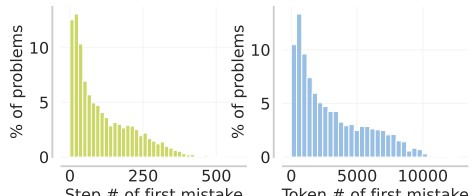

Figure 2: *Location of first occurring mistakes in incorrect rollouts.* Incorrect rollouts may contain many correct preceding steps before the mistake. Detected by Gemini 2.5 Pro with Prompt 3.

---

[1]For the LLMs we study, this decomposition is implemented by splitting the generated text using the delimiter ``\n\n'' (see Figure 3), following prior work (Qu et al., 2025b).

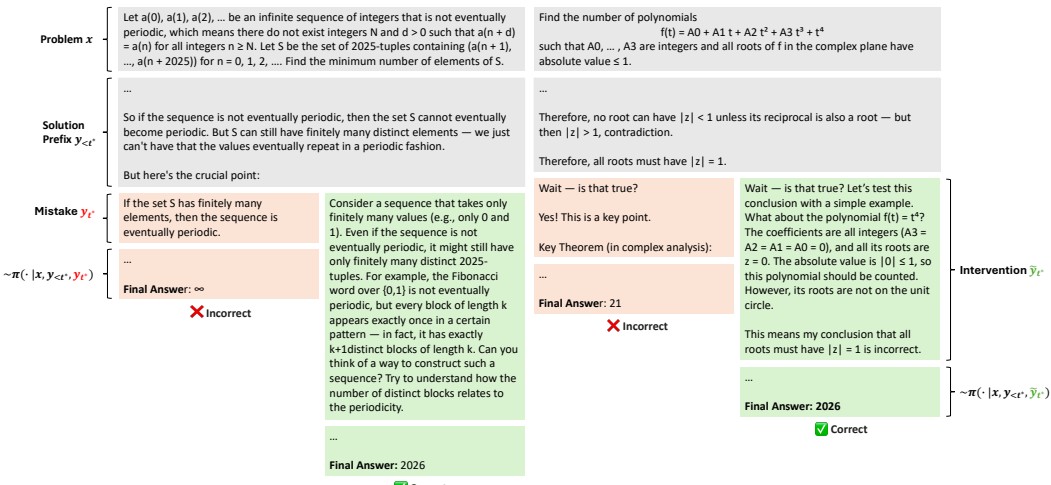

Figure 3: *Example mistakes encountered by Qwen3-4B-Instruct and proposed interventions.* By correcting the mistakes with interventions, the rollouts lead to correct final answers as opposed to originally incorrect ones.

## 3   CREDIT ASSIGNMENT VIA SELF-PROPOSED INTERVENTIONS

We develop a scalable approach for credit assignment in LLMs. We reduce this problem to two challenges: **(i)** identifying which reasoning steps are problematic and should receive low credit, and **(ii)** searching for alternative steps that attain higher credit. Most existing solutions to problem **(i)** rely on estimating value functions, which remains computationally expensive for LLMs. Approaches to problem **(ii)** typically require step-level on-policy RL; however, given the inherent difficulties of RL training for LLMs, these methods introduce an additional layer of complexity for practitioners. In this section, we provide an alternative approach to credit assignment that bypasses these challenges by leveraging the difficulty gap between instruction-following and problem-solving tasks.

**Main idea.** Our approach collapses the two steps above into a single procedure that directly proposes an improved reasoning step. While such a procedure might appear to require a stronger LLM, we show that meaningful corrective steps can be generated using the very same model. The key is to exploit the asymmetry in task difficulty between generating correct solutions from scratch and verifying individual steps by comparing against the reference solution. In practice, base models are often far more reliable at the latter: they can compare a generated reasoning trace against a known correct solution even when they cannot produce the solution themselves. Accordingly, we instruct the base model to use test-time compute to effectively perform a "diff" between a reasoning trace and a human-written reference solution, identify the step at which the trace goes astray, and propose a single-step corrective intervention. Next, we explain our approach for implementing this idea.

### 3.1   PROPOSING INTERVENTIONS THROUGH SELF-VERIFICATION

We leverage instruction-following LLMs to verify their own reasoning traces and propose counterfactual steps that lead to improved outcomes, circumventing the need for expensive rollouts or explicit training and optimization of PRMs. We implement this idea via a two-stage prompting procedure.

First, given a model-generated reasoning trace, we instruct the base model to verify each step of the trajectory for logical correctness, conditioned on access to a reference solution. This produces a list of reasoning steps that are potentially incorrect, which we require the model to index explicitly by their position in the trajectory. We then focus on proposing a corrective alternative for the *first* incorrect step. This design choice is motivated by the fact that the initial error must be corrected for an incorrect rollout to become correct. Moreover, errors that occur later in the trajectory are often consequences of earlier mistakes; once an early incorrect step is fixed, the remainder of the trajectory is unlikely to follow the same erroneous path, rendering corrections to later errors potentially unnecessary.

Upon identifying the first incorrect step $\mathbf{y}_{t^*}$ with step number $t^*$, we proceed to the second stage, in which the model generates an alternative next step $\tilde{\mathbf{y}}_{t^*}$ to replace $\mathbf{y}_{t^*}$. We refer to this alternative as an ***intervention***, whose purpose is to steer the remainder of the trajectory toward the correct answer. We provide concrete examples of generated single-step interventions in Figure 3, with additional examples in Appendix D and E. Concretely, we obtain the error location $t^*$ and corrective step $\tilde{\mathbf{y}}_{t^*}$ using a single query $\sim \pi(\cdot \mid \mathbf{x}, \mathbf{y}, \mathbf{p}_{\text{InT}})$; the instruction $\mathbf{p}_{\text{InT}}$ is provided in Box 2. Our

prompt borrows components from Huang & Yang (2025) to produce step-by-step verification logs of the trajectory and to identify the first mistake by classifying each step. Because $t^*$ and $\tilde{\mathbf{y}}_{t^*}$ are generated by the same model that produced the original $\mathbf{y}$, our approach enables credit assignment via *self-generated* feedback. Additionally, in the few cases where the intervention contained the final answer itself; we discard these cases to prevent any leakage from the reference solution, so that the model does not simply learn to memorize the final answer, but rather learns to complete the solution. Next, we evaluate whether interventions generated in this manner results in successful rollouts on problems that the base model could not solve at first.

## 3.2 Do Interventions Improve Success Rates?

We empirically evaluate self-generated interventions using the Qwen3-4B-Instruct-2507 base model by measuring whether a proposed intervention improves the probability of success on rollouts that would otherwise fail. Specifically, we concatenate problem $\mathbf{x}$ with reasoning steps preceding the identified error $\mathbf{y}_{<t^*}$, followed by the intervention $\tilde{\mathbf{y}}_{t^*}$. We then sample continuations from the model conditioned on this prefix, i.e., $\pi(\cdot \mid \mathbf{x}, \mathbf{y}_{<t^*}, \tilde{\mathbf{y}}_{t^*})$ (row 4 "4B-Instruct" in Table 4), and evaluate resulting trajectories to obtain an unbiased estimate of the expected reward. In Figure 4 (left), we isolate the effect of interventions by comparing this value against two baselines: **(1)** continuing the rollout from the erroneous step without applying the intervention $\pi(\cdot \mid \mathbf{x}, \mathbf{y}_{<t^*}, \mathbf{y}_{t^*})$ (row 3 above "4B-Instruct"), and **(2)** resampling from the step immediately preceding the error $\pi(\cdot \mid \mathbf{x}, \mathbf{y}_{<t^*})$ (row 2). Our results show that interventions increase the average reward from 0.0713% to 1.56%. This $22\times$ increase demonstrates that while the base model may be unable to solve these problems initially, it can leverage self-verification capabilities to generate single-step interventions $\tilde{\mathbf{y}}_{t^*}$ to replace the original erroneous step $\mathbf{y}_{t^*}$ and successfully steer the model toward a correct solution.

Beyond average reward, we also examine if utilizing our proposed interventions enables the model to solve new problems. To assess coverage, we compute the number of problems on which we can generate at least one correct rollout (out of 32) under each of the three settings in Figure 4 (right), which correspond to rows 2, 3, 4 in Table 4. Continuing from interventions achieves the highest coverage, solving 80 out of 334 unique problems compared to 29 out of 334 when continuing from the error. This justifies the efficacy of interventions, and further suggests that fine-tuning models to internalize interventions could potentially increase the fraction of problems receiving non-zero rewards during downstream RL, which would enable learning across a broader range of tasks. We include additional ablations and results in Appendix C.

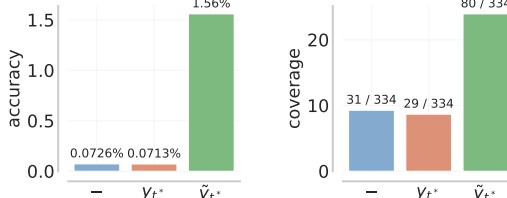

Figure 4: *Performance comparison of interventions.* Continuing the rollout by concatenating the proposed intervention step $\tilde{\mathbf{y}}_{t^*}$ to the prefix $\mathbf{y}_{<t^*}$ yields substantially higher accuracy (left) and a larger number of unique problems solved (right) than either continuing with the original erroneous step $\mathbf{y}_{t^*}$ or continuing directly from the prefix without any intervention $(-)$.

> **Takeaways: Effectiveness of our proposed interventions**
>
> - Conditioning rollouts on proposed interventions increases success rates by $\geq 20\times$.
> - Hint-guided rollouts are complementary to interventions. Both improve overall reward.
> - Instruction-following capability is critical: the instruction-tuned 4B-Instruct model solves 7 more problems (out of 334) than the reasoning-focused 4B model.
> - Intervention quality scales with model capacity: interventions proposed by 30B-A3B-Instruct achieve $2\times$ higher accuracy and solves more problems compared to 4B-Instruct.

## 4 INT: Intervention Training for Credit Assignment

Next, we use the generated interventions for training. Our training procedure is simple: we perform SFT on the collected intervention data to *patch* the base model, preparing it for subsequent RL training. Then RL training goes as usual, but now it can learn from these very challenging problems that would have otherwise largely produced incorrect rollouts during RL. As we show in our experiments, this patching step is sufficient to enable the model to solve previously unsolvable problems and leads to improvements in performance. We now describe our training procedure in detail.

### 4.1 Training via Supervised Fine-Tuning on Intervention Data

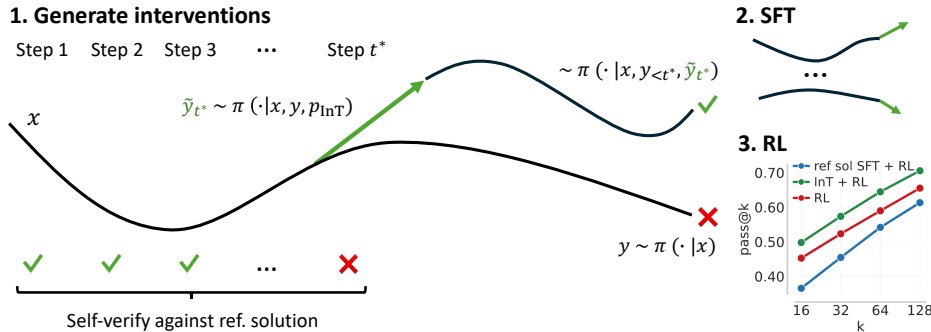

Figure 5: *Intervention training (`InT`).* `InT` verifies incorrect rollouts against reference solutions to identify the step $t^*$ at which the first error occurs, proposes alternative step $\tilde{\mathbf{y}}_{t^*}$, and performs SFT on these steps before RL.

A natural starting point for leveraging generated interventions during training is by running supervised fine-tuning (SFT) on the collected intervention data. The goal of this step is to teach the model to internalize the corrective patterns expressed by interventions, so that at test time, when reference solutions are no longer available, it can spontaneously generate similar corrective steps. This immediately raises the central design question surrounding this simple training procedure: **which parts of an intervention-guided rollout should be used in training?**

| Configuration | Coverage | Accuracy |
|---|---|---|
| Prefix, no suffix, filter | 202/235 | 7.71% |
| - no filter | 196/235 | 5.06% |
| - no prefix | 162/235 | 2.87% |
| + suffix | 111/235 | 2.31% |

Table 1: *A study of design choices for SFT on interventions.* We compare different SFT configurations: whether the prefix $\mathbf{y}_{<i}$ is cloned, the suffix $\mathbf{y}_{>i}$ is included, and interventions are filtered to retain only those that yield at least one correct rollout. Cloning the prefix and excluding the suffix while applying the correctness filter yields the highest coverage and accuracy. The 235 problems were subsampled from DeepScaleR.

Consider generating a full solution using an intervention. For a problem $\mathbf{x}$ and an incorrect response $\mathbf{y} \sim \pi(\cdot \mid \mathbf{x})$, we generate an intervention $\tilde{\mathbf{y}}_{t^*} \sim \pi(\cdot \mid \mathbf{x}, \mathbf{y}, \mathbf{p}_{\text{InT}})$ to replace the erroneous step $\mathbf{y}_{t^*}$. We then continue the response by sampling $\tilde{\mathbf{y}}_{>t^*} \sim \pi(\cdot \mid \mathbf{x}, \mathbf{y}_{<t^*}, \tilde{\mathbf{y}}_{t^*})$. Given the resulting intervention-guided trajectory, we consider several design choices for SFT: whether to train on the prefix $\mathbf{y}_{<t^*}$, the suffix $\tilde{\mathbf{y}}_{>t^*}$, and whether to include only the interventions whose continuations $\tilde{\mathbf{y}}_{>t^*}$ lead to a correct final answer.

**Main idea.** We hypothesize that cloning only the proposed intervention *without* its preceding prefix is insufficient. If the prefix $\mathbf{y}_{<t^*}$ is not reinforced during SFT, the fine-tuned model may generate alternative prefixes $\mathbf{y}'_{<t^*}$ for the same problem $\mathbf{x}$, introducing different errors than those encountered in the original rollouts used for training, rendering the intervention $\tilde{\mathbf{y}}_{t^*}$ irrelevant. While one could mitigate this issue by collecting multiple prefixes, doing so is impractical: covering the space of possible prefixes is already expensive, and *anticipating* which prefixes the fine-tuned model will produce is even more challenging as the output distribution $\pi(\cdot \mid \mathbf{x})$ shifts after fine-tuning. Motivated by these considerations, we follow Qu et al. (2024) and choose to clone both the prefix $\mathbf{y}_{<t^*}$ and intervention. We observe in our results in Table 1 that this choice leads to substantially improved coverage and accuracy, solving 40 more problems out of 235 compared to not cloning the prefix.

Next, we examine whether cloning the suffix $\tilde{\mathbf{y}}_{>t^*}$ is beneficial. We hypothesize that cloning a correct suffix could reduce the space of future sequences explored during subsequent RL training, thereby limiting exploration. Empirically, we find that this is indeed the case and cloning the suffix is detrimental. As shown in Table 1, training on full trajectories $(\mathbf{y}_{<t^*}, \tilde{\mathbf{y}}_{t^*}, \tilde{\mathbf{y}}_{>t^*})$ nearly halves the number of unique problems solved compared to training only on the prefix and intervention $(\mathbf{y}_{<t^*}, \tilde{\mathbf{y}}_{t^*})$. Finally, we observe that filtering interventions to retain only those that lead to at least one correct continuation over 32 rollouts, further improves post-SFT coverage by 6 out of 235 problems. **In summary,** the optimal SFT config consists of cloning the prefix and intervention, excluding the suffix, and applying a correctness filter. Our final SFT objective is given by:

$$\nabla_\pi J(\pi) \approx \mathbb{E}_{\mathbf{x} \sim D_{\text{train}}, \mathbf{y} \sim \tilde{\pi}(\cdot|\mathbf{x}) \text{ s.t. } r(\mathbf{x},\mathbf{y})=0} \left[ \nabla_\pi \log \pi(\tilde{\mathbf{y}}_{t^*} \mid \mathbf{y}_{<t^*}) + \sum_{t=0}^{t^*} \nabla_\pi \log \pi(\mathbf{y}_t \mid \mathbf{y}_{<t}) \right]. \quad (2)$$

## 4.2 FINE-TUNING VIA REINFORCEMENT LEARNING

After fine-tuning the base LLM on intervention data for a small number of steps, we proceed with RL post-training using standard outcome-reward methods such as GRPO (Shao et al., 2024). If the model has successfully "internalized" the intervention patterns, its rollouts should avoid

many of the errors observed in the base model while reinforcing previously correct behaviors. Starting RL from this SFT-initialized model then further amplifies these corrective behaviors, suppresses unproductive reasoning segments, and enables the extraction of learning signal from rollouts that previously yielded none. We refer to our approach of training LLMs on intervention data as `InT` (Intervention Training). Our practical approach shown in Algorithm 1 first patches the LLM on intervention data (Line 7) and then subsequently runs RL. We provide a schematic illustration in Figure 5.

---

**Algorithm 1** InT: Intervention Training

**Require:** Base LLM $\pi$, Intervention generation prompt $\mathbf{p}_{\text{InT}}$, Problems $\mathcal{D}_{\text{train}}$
1: **Data Collection:** $\mathcal{D}_{\text{InT}} \leftarrow \{\}$
2: **for** each $\mathbf{x} \in \mathcal{D}_{\text{train}}$ **do**
3:      Generate $\mathbf{y} \sim \pi(\cdot|\mathbf{x})$
4:      $t^*, \tilde{\mathbf{y}}_{t^*} \sim \pi(\cdot|\mathbf{x}, \mathbf{y}, \mathbf{p}_{\text{InT}})$ (error location, intervention)
5:      $\mathcal{D}_{\text{InT}} \leftarrow \mathcal{D}_{\text{InT}} \cup (\mathbf{x}, \text{concat}(\mathbf{y}_{<t^*}, \tilde{\mathbf{y}}_{t^*}))$
6: **end for**
7: **Patching:** $\pi' \leftarrow \text{SFT}(\pi, \mathcal{D}_{\text{InT}})$
8: **RL training:** $\pi'' \leftarrow \text{RL}(\pi', \mathcal{D}_{\text{train}})$
9: **return** $\pi''$

---

**Summary: Intervention Training (`InT`)**

Performing SFT on the proposed interventions with the prefix, without the suffix on filtered traces leads to the best performance and provides a good initialization for online RL.

## 5 EXPERIMENTAL EVALUATION OF INT

In our experiments, we compare models trained with `InT` against alternative approaches on challenging problems where standard RL training predominantly produces incorrect rollouts. Overall, we find that `InT` achieves the strongest performance across all settings - it reduces the fraction of problems that yield only incorrect rollouts during RL, consistently increases pass@$k$ across all evaluated values of $k$, and delivers strong gains on standardized benchmarks.

### 5.1 EXPERIMENTAL SETUP, COMPARISONS, AND EVALUATION METRICS

To construct a dataset of challenging problems, we draw from the following sources: **(1) Polaris** (53k problems), filtering for instances that achieve zero accuracy under 64 rollouts; **(2) AceReason-Math** (50k problems), similarly filtered for zero accuracy under 64 rollouts; **(3) Omni-MATH** (4.4k problems), filtered for zero accuracy under 128 rollouts; and **(4) IMO-AnswerBench** (360 problems), after removing those reserved for the test set. Reference solutions are available for most problems from Polaris, AceReason-Math (Luo et al., 2025), and Omni-MATH (Gao et al., 2024); for the remaining problems, we generate reference solutions using Gemini. After applying these difficulty filters, we obtain a pool of approximately 4500 hard problems. We then generate interventions for this set and retain only those that yield non-zero reward in at least one of 32 rollouts, resulting in 1076 problems with corresponding interventions. As shown in Table 4 (top), we further select a subset of 334 problems to compare naïve rollouts against rollouts with interventions.

**Baselines and comparisons.** To evaluate the efficacy of `InT`, we compare it against several alternative approaches for patching model behavior on $\mathcal{D}_{\text{train}}$, followed by running RL. Our primary comparisons are: **(1) Reference Solution SFT + RL**, which distills reference solutions into the base model prior to RL; **(2) Standard RL**, which applies RL directly to the base model; **(3) Self-Reflection SFT + RL**, where the model is prompted to rewrite entire solutions rather than propose single-step interventions (see Appendix H), followed by SFT and RL; and **(4) Hint-guided RL**, which runs RL on int-augmented problems (we use the dataset released by (Qu et al., 2025a), a hint-guided RL method targeting problems of comparable difficulty). For all RL runs, we run GRPO for 400 steps.

**Cost analysis of PRM training.** A natural comparison to our approach is to train a process reward model (PRM) and use step-level rewards derived from it. However, high-quality PRMs are currently not available for the Qwen3 models we use in this work. While prior studies have demonstrated that PRMs can be trained and utilized (Setlur et al., 2024b), doing so requires substantial training and data collection compute. For example, the Qwen2.5-Math-PRM series relied on roughly three million rollouts together with complex filtering pipelines. Additionally, recent

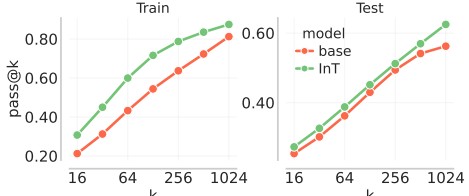

Figure 6: *Performance of the post-SFT model.* SFT on interventions increases pass@$k$ on both train and test splits for $k = 16, 32, \ldots, 1024$, on 32 randomly sampled problems from each split.

work suggests that existing PRMs tend to underperform on difficult problems (Feng et al., 2025). Given these challenges, we do not include a PRM-based comparison, and focus on alternative methods that leverage reference solutions to improve performance. Compared to PRM-based approaches, `InT` provides a much simpler and more scalable mechanism for credit assignment.

**Evaluation benchmarks and metrics.** Prior work on RL-trained reasoning models primarily evaluates performance on intermediate competition math benchmarks such as AIME 2025 and HMMT 2025. However, these benchmarks are already quite old compared to release dates of current base models. To address this limitation, we evaluate `InT` on substantially harder standardized benchmarks. Specifically, we report results on IMO-AnswerBench (Luong et al., 2025), AMO-Bench (An et al., 2025), and Apex Shortlist (Balunović et al., 2025), all of which consist of Olympiad-level problems released after the public release of our base model, Qwen3-4B-Instruct-2507. In addition, we scrape the November 2025 HMMT competition from the official HMMT website to evaluate on problems constructed after the release of the base model, further reducing the risk of training-set contamination. For IMO-AnswerBench, we manually select 40 problems: 10 each from Algebra, Combinatorics, Geometry, and Number Theory, restricting to problems with easily verifiable answers (e.g., integers or simple fractions) to ensure that evaluation reflects mathematical reasoning rather than sensitivity to answer parsing. We apply an analogous filter to AMO-Bench, removing problems whose answers cannot be verified automatically (e.g., proof-style responses).

## 5.2 RESULTS: SFT WITH INTERVENTIONS

We examine the performance of `InT` immediately after SFT on intervention data (Section 4; "patching"). Prior work suggests that RL in LLMs primarily reallocates probability mass toward already-existing successful trajectories, effectively converting high pass@$k$ in a base model into high pass@1 after RL training (Yu, 2025; He et al., 2025; Yue et al., 2025). From this perspective, pass@$k$ serves both as a diagnostic of whether the model is capable of sampling correct solutions and as an upper bound on downstream accuracy achievable through subsequent RL training. Motivated by this view, we compare the pass@$k$ of the SFT model to that of the base model on both the training and test sets in Figure 6. The SFT model consistently achieves higher pass@$k$ across $k = 16, 32, \ldots, 1024$ on both splits, indicating that fine-tuning on intervention data enriches the distribution of successful trajectories and provides a strong initialization for

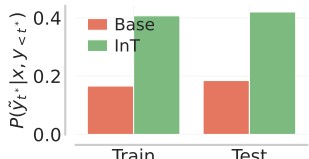

Figure 7: *Average probability of intervention tokens.* Intervention tokens receive higher probability under the trained model vs. the base model on both the training and test sets, indicating that SFT causes the model to learn to produce intervention-like steps even on unseen test problems.

RL. In the next section, we show that further RL training from this checkpoint indeed translates these gains into improved final performance. We also find that likelihood of intervention tokens increases post-SFT (Figure 7), indicating that `InT` enables the model to internalize interventions successfully.

Next, we compare `InT` against alternative approaches to generate corrections to incorrect rollouts. Specifically, we evaluate performance after running SFT on **Self-Reflection** traces, **Reference** solutions, and thinking traces from DeepSeek-R1 (**R1 Think** and **R1 Summary**, which correspond to content inside and after the `<think>` tags, respectively), to compare `InT` with fine-tuning on traces generated from a more capable LLM. Across both training and test sets, we find that `InT` consistently improves pass@1 and pass@$k$ relative to all comparisons in Figure 9(train middle, test right).

**Finding: `InT` traces are more likely under the base model.** Beyond improvements in pass@$k$, `InT` generates traces for SFT that yield the highest likelihood under the base model. Figure 9(left) reports the average negative log-likelihood (NLL) of tokens drawn from trajectories generated by different sources, including human-written reference solutions, Gemini-generated reference solutions, DeepSeek-R1 outputs, as well as self-reflection and `InT` trajectories. We find that `InT` trajectories are the most likely under the base model distribution, followed by self-reflection, R1 outputs, and reference solutions. This observation is important because fine-tuning on highly off-policy traces is known to be problematic. Prior work (Kang et al., 2024) shows that such traces are often memorized by the model, and fitting them can significantly distort the base model's next-token distribution. Consistent with this perspective,

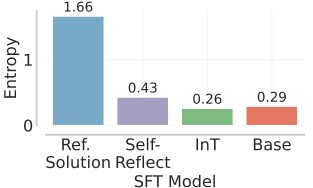

Figure 8: **Entropy of the next-token distribution** for various models. SFT on off-policy traces leads to elevated entropy, while `InT` preserves a low-entropy distribution closer to the base model, yielding more stable initialization for subsequent RL.

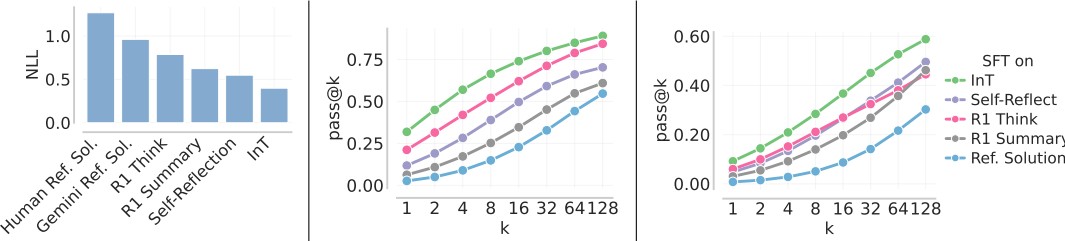

Figure 9: **`InT` trains on tokens that are more likely under the base model and attains highest performance on both train and test problems. (left)** Average negative log-likelihood (NLL) computed over 64 sampled traces. **Ref. Sol.** denotes reference solutions written by humans or the Gemini-2.5-Pro model. **R1 Think** and **R1 Summary** correspond to content inside and after the DeepSeek-R1 `<think>` tags, respectively. **Self-reflect** refers to the self-reflection baseline in which the model is prompted to rewrite entire incorrect solutions given a reference solution. `InT` trains on the most on-policy traces as it exhibits the lowest NLL. **(middle)** Train pass@$k$ on 64 sampled training problems. `InT` achieves the highest pass@$k$ across all values of $k$. **(right)** Test pass@$k$ on IMO-Bench, AMO-Bench, and Apex Shortlist. `InT` again attains the best performance across all $k$.

Figures 9(middle, right) show that the relative performance of different methods closely tracks the ordering of their NLL values. While this correlation is not necessarily causal, it motivates a closer examination of why highly off-policy traces pose challenges for fine-tuning.

***Why does cloning off-policy traces perform poorly?*** Fine-tuning on highly off-policy traces requires shifting probability mass away from tokens with high likelihood under the base model toward tokens it considers unlikely. This redistribution naturally increases the entropy of the next-token distribution, an effect we observe empirically in Figure 8. Such high-entropy initializations are problematic for subsequent RL training, as they lead to overly random rollouts that hinder effective exploration and learning. One might attempt to mitigate this issue by training for longer, allowing the model to eventually assign higher likelihood to the off-policy tokens. However, this introduces a different failure mode: prolonged fine-tuning on off-policy data (e.g., human-written reference solutions) distorts the base model's existing reasoning patterns, since these traces differ substantially from the behaviors and skills the model has already learned. As a result, cloning off-policy trajectories performs poorly regardless of whether training is stopped early or run for longer, and might not produce good RL initializations.

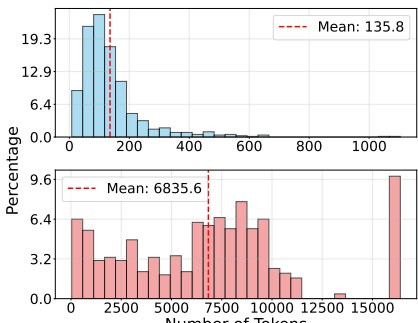

Figure 10: ***Interventions are short.*** **Top:** Intervention are typically under 200 tokens. **Bottom:** full rollouts are $\sim$ 7k tokens on avg.

In contrast, `InT` generates traces with most tokens coming from the base model because interventions are short compared to full rollouts 10, leading to trajectories that with higher likelihood under the base model. This avoids large shifts in entropy, as shown in Figure 8: the resulting entropy of the next-token distribution after applying `InT` is roughly comparable to that of the base model and can thus support effective exploration during subsequent RL training.

## 5.3 RESULTS: INT + RL SOLVES MOST HARD TRAINING PROBLEMS

Next, we show that on our curated training set of difficult problems, initializing RL from the patched checkpoint produced by `InT` leads to the largest increase in reward and the greatest reduction in the fraction of problems that produce no correct rollouts (the "zero-advantage ratio"), as shown in Figure 11. This result confirms that training on interventions can effectively help avoid errors made by the base model during RL. Moreover, we observe a similar effect on test problems. On two unseen problems selected from the IMO shortlist, `InT` is able to correct errors made by the base model, as illustrated qualitatively in Appendix I.

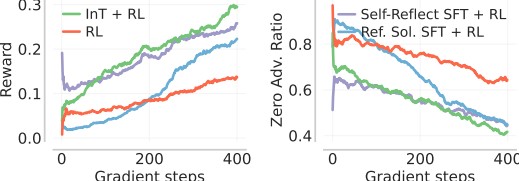

Figure 11: ***RL training stats.*** **Left**: RL training reward. **Right**: Zero-advantage ratio - fraction of problems for which the model never produces a correct rollout. `InT` yields both higher reward and a substantially lower zero-advantage ratio, enabling learning on problems that previously yielded no signal. The next best approach is Self-Reflection SFT + RL, but it leads to much weaker test performance (see Table 3).

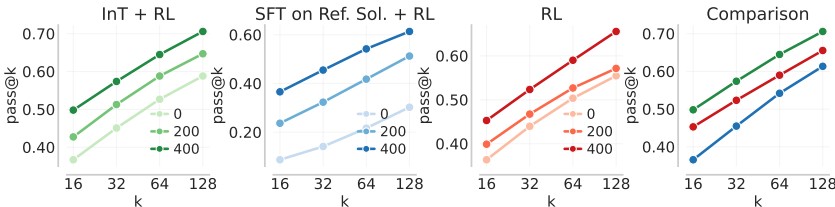

Figure 12: **Pass@$k$ across RL training iterations on test problems.** We plot pass@$k$ for three model initializations: (i) base model patched with InT, (ii) base model fine-tuned on reference solutions, and (iii) base model itself. Across all values of $k$, InT achieves the highest pass@$k$ through training (right).

## 5.4 RESULTS: InT EXPANDS THE PASS@$k$ FRONTIER ON TEST PROBLEMS

We next present the main results of InT on the full evaluation set described in Section 5.1. Specifically, we track pass@$k$ performance across RL training iterations from 0 to 400 for three models: **(i)** the base Qwen3-4B-Instruct model, **(ii)** Qwen3-4B-Instruct fine-tuned on reference solutions, and **(iii)** Qwen3-4B-Instruct patched with InT. We find that initializing RL from the model produced by InT leads to the largest improvements in pass@$k$ throughout training. As summarized in Figure 12, InT outperforms both the base model and reference solution SFT across RL iterations.

## 5.5 RESULTS: PERFORMANCE ON STANDARDIZED BENCHMARKS

In the previous section, we showed that InT pushes the pass@$k$ frontier across all values of $k$ on both training and test problems. We now examine whether these gains translate into improved pass@1 performance on held-out benchmarks. To this end, we evaluate InT on four challenging reasoning benchmarks: IMO-AnswerBench, HMMT 2025 (November), AMO-Bench, and Apex Shortlist.

As shown in Table 3, InT achieves an average score of 33.72 across these benchmarks, corresponding to a ~59% relative improvement over the base model (21.17) and a ~19% improvement over the SFT+RL baseline (28.26). On IMO-AnswerBench, a high-quality test set curated by IMO medalists, InT attains a score of 25.62, more than double the base model performance (11.68) and surpassing standard RL (23.46).

| Model | IMO-AnswerBench |
|---|---|
| Qwen3-4B-Instruct-2507 | 11.68% (16K) |
| DeepSeek-R1-0528-Qwen3-8B | 18.44% (32K) |
| gpt-oss-20b | 23.36% (32K) |
| **Qwen3-4B-Instruct-2507 + InT + RL (Ours)** | 25.62% (16K) |

Table 2: *Evaluation on IMO-AnswerBench.* Performed at different "official" token budgets.

Moreover, we find that performing SFT on the reference solution does not synergize well with subsequent RL, attaining an average score of 20.76 that is *lower* than that of the untuned base model (21.17). This suggests that naïvely fine-tuning on off-policy reference solutions may restrict effective exploration in downstream RL, as discussed in Section 5.2. In contrast, by remaining on-policy, InT enables RL to further improve performance, as reflected by consistent gains across all metrics. While Self-Reflection SFT shows promise on certain benchmarks (e.g., 36.72 on AMO-Bench), it generalizes less robustly than InT. On IMO-AnswerBench, Self-Reflection attains only 15.53, compared to 25.62 for InT. InT achieves the highest reward value of 28.83, boosting the training reward higher than any other method while not falling into memorization traps of off-policy SFT.

| Model | IMO-AnswerBench | HMMT 2025 Nov | AMO-Bench pass@8 | Apex Shortlist pass@8 | $D_{\text{train}}$ | Average |
|---|---|---|---|---|---|---|
| Base | 11.68 | 41.61 | 26.24 | 20.79 | 5.53 | 21.17 |
| + RL | 23.46 | 46.46 | 35.21 | 22.72 | 13.47 | 28.26 |
| + Hint-guided RL | 16.89 | 47.27 | 33.34 | 22.23 | 23.06 | 28.56 |
| + SFT on ref. solutions + RL | 11.56 | 27.45 | 25.19 | 20.51 | 19.07 | 20.76 |
| + SFT on self-reflections + RL | 15.53 | 38.65 | **36.72** | 23.93 | 23.19 | 27.60 |
| **+ InT + RL (Ours)** | **25.62** | **49.77** | 36.16 | **28.22** | **28.83** | **33.72** |

Table 3: **Performance on challenging math benchmarks.** Pass@1 (IMO-AnswerBench, HMMT 2025 Nov) and pass@8 (AMO-Bench, Apex Shortlist) estimated using 128 rollouts. $D_{\text{train}}$ reports the average train reward at step 400, smoothed using exponential moving average. InT achieves the strongest overall performance followed by hint-guided RL, standard RL, and other SFT approaches.

> **Summary: Intervention Training (InT) Results**
> - More "off-policy" SFT data leads to higher entropy and worse pass@$k$ performance
> - InT leads to the greatest increase in training reward and decrease in zero-adv ratio.
> - InT achieves the highest test accuracy on a variety of difficult benchmarks, including a nearly 14% increase in IMO-AnswerBench, surpassing gpt-oss-20b with a 4B model.

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

ACKNOWLEDGEMENTS

We thank Bhavya Agrawalla, Taeyoun Kim, Yuxiao Qu, Zheyuan Hu, Baolong Bi, Jiahe Jin, and the rest of the AIRe lab at CMU for informative discussions, feedback and input on our results and a previous version of this paper. This research is supported by a gift from Apple, Schmidt Sciences AI2050 Early Career Fellowship, and an Open Philanthropy grant on technical AI safety.

This research used the DeltaAI advanced computing and data resource, which is supported by the National Science Foundation (award OAC 2320345) and the State of Illinois. DeltaAI is a joint effort of the University of Illinois Urbana-Champaign and its National Center for Supercomputing Applications. We thank the staff at DeltaAI for their continual and timely assistance with compute allocation. Experiments in this paper also utilized H100 GPU resources from the Orchard cluster in the FLAME center at CMU for which we are grateful for. AK also thanks the TRC program at Google Cloud for their continued support.

## A  RELATED WORK

**Credit assignment in LLM reasoning.** The effectiveness of long-horizon RL with outcome rewards (DeepSeek-AI et al., 2025; Jaech et al., 2024; Team et al., 2025; Hugging Face, 2025) is often crippled by credit assignment: it is unclear which intermediate steps in a long response should be "credited" for the outcome reward. Credit assignment is usually not a problem when responses are short and the model can produce enough successful rollouts to contrast them against incorrect ones, but can pose as significant bottlenecks when the horizon increases (Setlur et al., 2025a; Wu et al., 2025b; Gan et al., 2025). While most methods reward each token with the outcome-level advantage (Yu et al., 2025; Zeng et al., 2025; Liu et al., 2025b), others use process reward models (PRMs) to assign dense token or step-level rewards (Lightman et al., 2023; Wang et al., 2024; Qu et al., 2025b) that can reinforce correct steps and promote unlearning of incorrect ones. Although PRMs may improve RL compute efficiency (Setlur et al., 2024b;a), the notion of what a process reward should measure and how to calculate it precisely is not yet settled. Some work derives process rewards by running "branched" rollouts which is costly (Luo et al., 2024; Setlur et al., 2024b; Kazemnejad et al., 2024), while other work trains PRMs with human annotations and suffers from reward hacking (Cheng et al., 2025; Wang et al., 2025a). Even after training a PRM, one must still devise mechanisms to optimize it at a per-step level in order to identify alternative steps that lie on a correct trajectory, a task that is itself nontrivial (Wang et al., 2025a; Zhang et al., 2025c). In contrast, our work leverages the asymmetry between a model's instruction-following and solution generation capabilities to enable direct verification of a reasoning trace and construction of a corrective step that redirects the trajectory. This approach effectively amortizes value estimation and policy optimization into a single procedure, resulting in a simpler and cheaper approach for credit assignment.

While our approach, in principle, can be viewed as using generative models for some form of verification (Zhang et al., 2024; Liu et al., 2025c; Kim et al., 2025; Khalifa et al., 2025), it is distinct from these prior works in that we task the model with explicitly pointing out the location of a single, key mistake, rather than verifying every step and judging the solution to produce a binary judgement of outcome-level or process-level correctness. We amortize value estimation with policy optimization to directly produce an improved step (intervention). Moreover, unlike the majority of work on generative reward models or verifiers, our focus is on improve training directly rather than using inference-time methods (e.g., best-of-$N$ search).

**Learning from natural language feedback.** A related line of work explores using natural language feedback to improve RL training. These approaches typically leverage feedback to refine rollouts, which are then used to update the policy. For example, Chen et al. (2024) combine human feedback with a separate refinement model to improve policy-generated outputs that are distilled back into the policy via SFT. Yan et al. (2025) use a teacher model to generate correction trajectories for off-policy RL, while Zhang et al. (2025a) employ critique-guided self-refinement to generate improved trajectories for off-policy RL. In contrast, our approach generates short, targeted natural language feedback to correct individual steps within otherwise on-policy trajectories. Moreover, interventions are proposed by the same model that produced the incorrect reasoning trace, without relying on human feedback or stronger external models.

Additionally, there have been a class of "text gradient" methods (Lee et al., 2025; Madaan et al., 2023; Yuksekgonul et al., 2024) that leverage LLMs to generate textual feedback to improve test-time

performance. While our method is similar in that it also employs a textual learning signal, it differs in two key ways: the feedback is short and targeted, and it is used to improve training rather than solely to enhance inference-time behavior. Finally, as we show in Section 3, the use of a reference solution substantially improves performance compared to approaches that only rely on the priors in the base model to generate corrective feedback.

**Hint-guided RL.** Several concurrent approaches also leverage reference solutions in RL, but differ in how this information is used. Rather than using reference solutions for verification and credit assignment, these methods guide the policy directly. For example, (Li et al., 2025; Amani et al., 2025; Zhang et al., 2025b; Qu et al., 2025a) incorporate expert-generated guidance in the form of *partial-solution prefixes*, i.e., static rationale snippets that the model conditions on during training. While such approaches substantially improve exploration by enabling the model to sample successful trajectories on otherwise unsolvable problems, they do not directly address credit assignment on incorrect traces. As a result, our method is complementary to this line of work and can be naturally combined with it, a synergy we demonstrate empirically in Section 3.

Moreover, we find in Figure 2 that for a substantial fraction of problems, the first incorrect step occurs deep into the reasoning trace. In particular, over 70% of errors appear after the first 50 steps. Existing hint-guided methods typically repurpose a prefix of a reference solution as a hint, and cannot directly tackle errors that arise later in a trace. One could attempt to mitigate this by providing longer hints, but this approach risks enabling the model to simply rely on the hint itself, effectively learning to "cheat".

**Training with interventions outside of LLMs.** Applying interventions at points of failure has been explored in domains beyond LLM training. For example, DAgger (Ross et al., 2011) mitigates compounding errors in imitation learning by querying an expert for labels on states visited by the learner's own policy. HG-DAgger (Kelly et al., 2019) adapts this framework to self-driving, and subsequent work extends it to robotic manipulation and long-horizon control tasks (Hu et al., 2025; Wu et al., 2025a; Liu et al., 2025a; Mandlekar et al., 2020). Across these domains, intervention-based methods typically achieve improved data efficiency and faster convergence compared to behavior cloning, due to more effective credit assignment. In contrast to this line of sprior work, where interventions are provided by humans, we show that a model can self-propose interventions to correct its own reasoning given a reference solution, by leveraging asymmetries in its capabilities.

## B    DISCUSSION AND PERSPECTIVES ON FUTURE WORK

In this work, we introduce `InT`, a simple yet effective approach for assigning credit to incorrect rollouts. `InT` addresses this challenge by instructing the model to self-verify and propose single-step corrective interventions. We then fine-tune the model on these constructed interventions and perform RL post-training on the resulting patched model. By enabling more precise credit assignment, `InT` achieves strong performance in both training and test settings, allowing off-policy reference solutions to improve already capable models while avoiding the typical drawbacks of off-policy supervised fine-tuning. Looking ahead, several promising directions emerge from this work:

**Self-improvement by combining different LLM capabilities.** In this work, we leverage the asymmetry between generation and verification capabilities of LLMs to assign credit to incorrect rollouts. A natural next step is to further strengthen verification by explicitly training models on verification tasks, potentially eliminating the reliance on reference solutions. This would enable more autonomous self-improvement by removing the dependency on human-provided solutions. Moreover, stronger verification would allow LLMs to tackle tasks that require step-by-step correctness checking, such as IMO-ProofBench Luong et al. (2025). Ideally, verifiers would be trained in a continual fashion, improving alongside the solution generation model. Concurrent work by Shao et al. (2025) explores this direction by training a meta-verifier that validates verifications themselves, but training this meta-verifier itself needs human or bigger model supervision. Beyond verification, another key capability for self-improvement is problem generation. For example, could LLMs learn to construct their own training problems by modeling trajectories encountered by expert problem proposers, such as those in IMO-style competitions? If these components can be combined, they suggest a fully autonomous system in which one model proposes problems, another solves them, and a verifier provides reward signals that drive continual parameter updates.

**Credit assignment in continual learning.** In this work, we study credit assignment in a single-turn setting for solving math problems. An important direction for future work is to extend these

| Intervention Proposed By | Rollouts Are Conditioned On | Coverage | Accuracy |
|---|---|---|---|
| *Omni-MATH subset with pass@128 = 0, values estimated with 32 rollouts* | | | |
| 1. - | problem $\mathbf{x}$ | 40 / 334 | 0.0984% |
| 2. - | problem $\mathbf{x}$ + correct prefix $\mathbf{y}_{<t^*}$ | 31 / 334 | 0.0726% |
| 3. - | problem $\mathbf{x}$ + correct prefix $\mathbf{y}_{<t^*}$ + original step $\mathbf{y}_{t^*}$ | 29 / 334 | 0.0713% |
| 4. 4B-Instruct (**Ours**) | problem $\mathbf{x}$ + correct prefix $\mathbf{y}_{<t^*}$ + intervention $\tilde{\mathbf{y}}_{t^*}$ | 80 / 334 | 1.56% |
| 5. 4B (**Ours**) | problem $\mathbf{x}$ + correct prefix $\mathbf{y}_{<t^*}$ + intervention $\tilde{\mathbf{y}}_{t^*}$ | 73 / 334 | 2.65% |
| 6. 30B-A3B-Instruct (**Ours**) | problem $\mathbf{x}$ + correct prefix $\mathbf{y}_{<t^*}$ + intervention $\tilde{\mathbf{y}}_{t^*}$ | 101 / 334 | 2.87% |
| 7. 4B-Instruct w/o. ref. sol. | problem $\mathbf{x}$ + correct prefix $\mathbf{y}_{<t^*}$ + intervention $\tilde{\mathbf{y}}_{t^*}$ | 47 / 334 | 0.171% |
| *Subset with hint-guided pass@16 = 0, values estimated with 16 rollouts* | | | |
| 8. - | problem $\mathbf{x}$ + hint $\mathbf{h}$ + correct prefix $\mathbf{y}_{<t^*}$ | 11 / 176 | 2.38% |
| 9. 4B-Instruct (**Ours**) | problem $\mathbf{x}$ + correct prefix $\mathbf{y}_{<t^*}$ + intervention $\tilde{\mathbf{y}}_{t^*}$ | 18 / 176 | 3.05% |
| 10. 4B-Instruct | problem $\mathbf{x}$ + hint $\mathbf{h}$ + correct prefix $\mathbf{y}_{<t^*}$ + intervention $\tilde{\mathbf{y}}_{t^*}$ | 25 / 176 | 4.62% |
| *Final Training Set* | | | |
| 11. 4B-Instruct (**Ours**) | problem $\mathbf{x}$ + correct prefix $\mathbf{y}_{<t^*}$ + intervention $\tilde{\mathbf{y}}_{t^*}$ | 1076 / 4048 | 4.27% |

Table 4: **Rollouts conditioned on proposed interventions and alternative strategies.** Accuracy measures the average reward, and coverage measures the number of problems that see at least one correct rollout. Conditioning rollouts on our proposed interventions yields the highest coverage and average reward on the hard problem set, outperforming approaches that do not use interventions (e.g., naïve rollouts or conditioning on hints) and those that generate interventions under weaker supervision (e.g., without reference solutions). We also find that our interventions can be combined with existing approaches such as hint-guided rollouts (see rows 8-10) to obtain stronger results. 4B-Instruct denotes Qwen3-4B-Instruct-2507, 4B denotes Qwen3-4B, and 30B-A3B denotes Qwen3-30B-A3B-Instruct-2507.

ideas to continual improvement regimes, where rollouts are progressively compressed into LLM-generated summaries and or neural memory modules, and the effective context evolves over time. For example, how can credit be traced back to earlier decisions that may persist only through memory representations? How can we disentangle errors arising from poor generation from those caused by imperfect memory, and jointly improve memory architectures alongside the generation model? These questions are interesting to study in future work.

## C  ABLATIONS ON DIFFERENT WAYS OF PROPOSING INTERVENTIONS

In this section, we study ablations that isolate which ingredients are most important for proposing effective interventions. Overall, interventions provide a more fine-grained mechanism for steering rollouts than hint-only conditioning, and their effectiveness depends on access to reference solutions, strong instruction-following behavior, and model capacity.

**Ablation: Interventions outperform hint-guided rollout generation.** Having established that interventions improve reward and enable solving problems that were previously unsolvable, we now compare this approach to alternative ways of leveraging reference solutions. As discussed in Section A, several concurrent works incorporate reference solutions into RL by conditioning the policy on partial solution prefixes as *hints*. These hints exploit the instruction-following capabilities of base models to "steer" generation toward the correct region of the solution space, enabling successful completions on otherwise difficult problems. Since hints cannot directly reveal the solution, we construct them by randomly sampling a prefix from the initial one-third of the reference solution. We study whether hint-guided rollouts are more effective than intervention-based rollouts, and whether the two approaches can be combined.

Table 4 compares these settings. Conditioning on interventions alone (row 9; $\pi(\cdot \mid \mathbf{x}, \mathbf{y}_{<t^*}, \tilde{\mathbf{y}}_{t^*})$) achieves higher problem coverage (18/176 problems solved) than hint-guided rollouts (row 8) that solves only 11/176 problems. However, the two methods are complementary: conditioning on both hints and interventions (row 10) yields the best overall coverage (25/176 problems solved). This result is perhaps natural: hints guide the model toward a promising initial direction, while interventions enable fine-grained credit assignment when errors arise in the middle of a rollout.

**Ablation: Reference solutions improve effectiveness of interventions.** We next isolate the role of reference solutions in generating effective interventions. In particular, we test the alternative hypothesis that the verification–generation asymmetry of the base model alone (Setlur et al., 2025a) is sufficient to produce high-quality interventions. To do so, we evaluate an ablation shown in Table 4, row 7, where we prompt the model to propose an intervention given only an incorrect reasoning

trace, without the reference solution. On the one hand, conditioning on interventions outperforms the naïve baselines (rows 1–3), doubling accuracy and increasing coverage by 7, supporting the claim that effective interventions can be yielded even without privileged information from reference solutions. On the other hand, interventions generated in this manner are significantly weaker than those generated with reference solutions (row 4), which clearly highlights the critical role of reference solutions in producing high-quality interventions.

**Ablation: Instruction-following capabilities of the base model are critical.** We find that the instruction-following ability of the base model is a key determinant of intervention effectiveness. In row 5 of Table 4, we evaluate Qwen3-4B, a model optimized primarily for *reasoning* rather than instruction-following. Despite its strong performance on math reasoning benchmarks, this model frequently fails to adhere to the instructions required for intervention generation. For example, it may switch to directly solving the problem midway through comparing the reasoning trace with the reference solution, or neglect to enclose proposed interventions within the required <intervention></intervention> tags. Such instruction-following failures prevent the model from producing usable interventions, resulting in lower coverage (73 out of 334, row 5) compared to self-generated interventions from the instruction-tuned model (80 out of 334, row 4). These results demonstrate that effective intervention generation depends not only on raw reasoning ability, but also on robust instruction-following abilities.

**Ablation: More capable base models propose better interventions.** Finally, we study how the size of the model used to propose interventions affects performance. In addition to using the base model, Qwen3-4B-Instruct-2507, which also generates the original reasoning traces, we experiment with a larger model from the same family, Qwen3-30B-A3B-Instruct-2507 (row 6), to generate interventions. Using the larger model yields an approximately $2\times$ improvement in accuracy and solves 21 additional problems out of 334, indicating that the quality of generated interventions scales with model capacity.

# D PROMPTS

---

**Box 1: Prompt for intervention self-generation**

You are an expert mathematician teaching a Math Olympiad class. You will be given a problem and a high-level reference solution to the problem. Your task is to solve the problem step-by-step guided by the high-level reference solution.

**Problem**
{insert problem}

**High-Level Reference Solution**
{insert reference solution}

Great job! Now, a student in your class has solved the problem incorrectly. You must leverage your understanding of the reference solution to help him correct his attempt at the problem.

**Detailed Instructions**

**1. Detailed Verification Log**

You must perform a **step-by-step** check of the student's attempt against the reference solution. The steps here refer to each numbered substep in the **Incorrect Student Attempt** (i.e., **Substep 1:**, **Substep 2:**, . . . ), and not the high-level steps (e.g., **Step 1:**) which may contain multiple numbered substeps.

This analysis will be presented in a **Detailed Verification Log**, where you justify your assessment of each substep:

- For correct substeps, a brief justification suffices.
- For substeps with errors or gaps, you must provide a detailed explanation.

**Be careful and check every intermediate result, they are very easy to miss.**

**2. Identify the First Critical Error**

For each issue in the detailed verification log, you MUST determine whether it is a **critical error**. A critical error must pass the following two checks:

1. A critical error is either a **factual error** (e.g., a calculation error like `2+3=6`) or a **logical fallacy** (e.g., claiming that `A>B, C>D` implies `A-C>B-D`) that disrupts the current line of reasoning.
   **Procedure:**
     - Explain the specific error.
     - State explicitly that it **invalidates the current line of reasoning**.

---

2. A critical error must not be recovered from.

   **Procedure:**

   - Double-check that the error is not corrected or disputed in later steps (e.g., statements such as "Wait, but let me double-check this claim...").

As long as the issue passes both checks above, it is considered a **critical error**. We are interested in the *first* critical error that the student makes.

### 3. Propose an Intervention Step

After finding the critical error, you should propose an **intervention step** that will be inserted before the occurrence of the critical error to steer the student towards the correct solution.

- The intervention must **not give away the exact answer**.
- The student should still be able to learn from the error.
- You should provide guidance by **sketching the next steps** that lead toward the correct solution.

### 4. Output Format

Your response MUST be structured into three main sections:

1. **Detailed Verification Log**
2. **Critical Error Report**
3. **Intervention Step**

### 4.1 Detailed Verification Log

Provide the full, step-by-step verification log as defined above, structured in bullet points. When referring to the student's attempt or the reference solution, **quote the relevant text** before providing analysis.

### 4.2 Critical Error Report

In this report, include a bulleted list summarizing **every** issue discovered. For each issue, include:

- **Location:** A direct quote of the key phrase or equation where the issue occurs.
- **Issue:** A brief description of the problem and whether it is a **Critical Error**.

Stop once the **first** critical error has been found.

**4.3 Intervention Step**

Provide a single intervention step to be inserted **before** the critical error. The content must be written **from the student's perspective**. If there are no critical errors, leave this section empty.

**Draft of the Intervention Step Format**

1. **Content:** The text of the intervention step.
2. **Location:** The substep number where it should be inserted, including a quote from the student's attempt.

**Self-check**

Answer each question below with:

- A brief explanation, and
- A final verdict: Yes or No.

1. Does the **content** of the intervention step include the exact answer from the reference solution (i.e., "{correct_answer}")?
2. Is the **content** of the intervention step missing any key insights necessary to solve the problem?
3. Is the **location** of the intervention step later than the substep containing the first critical error?

If the answer to **any** question is "Yes", the draft has **failed the self-check**. Revise and repeat until all answers are "No".

If all answers are "No", write the final intervention step using the format below.

**Final Intervention Step Format**

INSERT_STEP_CONTENT should be the content of the intervention step, and INSERT_STEP_NUMBER should be an integer indicating the insertion substep.

<intervention>
{{ "content": INSERT_STEP_CONTENT, "location": INSERT_STEP_NUMBER }}
</intervention>

**Incorrect Student Attempt**
{student solution}

**Box 2: Prompt for Gemini intervention**

{insert problem}

{insert reference solution}

You have solved the problem correctly. Now, a student in your class has attempted the same problem. Your task now is to go over his solution step-by-step and write down a **detailed verification log**, identify the first **critical error**, and suggest locations in his solution to insert a replacement step such that if he follows the replacement step, it will guide him away from the error. Detailed instructions are listed below.

**Detailed Instructions**

**1. Detailed Verification Log**

You must perform a **step-by-step** check of the entire solution. This analysis will be presented in a **Detailed Verification Log**, where you justify your assessment of each step:

- For correct steps, a brief justification suffices.
- For steps with errors or gaps, you must provide a detailed explanation.

**Please be careful and check every intermediate result, they are very easy to miss.**

**2. Identify the First Critical Error**

For each issue in the detailed verification log, you MUST determine whether it is a **critical error**. A critical error must pass the following two checks:

1. A critical error is either a **factual error** (e.g., a calculation error like `2+3=6`) or a **logical fallacy** (e.g., claiming that `A>B, C>D` implies `A-C>B-D`) that disrupts the current line of reasoning.
   **Procedure:**
   - Explain the specific error.
   - State explicitly that it **invalidates the current line of reasoning**.
2. A critical error must not be recovered from.
   **Procedure:**
   - Double-check that the error is not corrected or disputed in later steps (e.g., statements such as "Wait, but let me double-check this claim...").

As long as the issue passes both checks above, it is considered a **critical error**. We are interested in the *first* critical error that the student makes.

**3. Propose Replacement Steps**

After finding the critical error, you must now identify existing steps in the student's solution that you can rephrase such that if the student were to begin from your rewritten step, he will be guided away from the critical error.

Note that replacement steps can occur either:

- **Before** the error, to circumvent it completely, or
- **After** the error, to recognize the mistake, dispute it, and recover with a correct argument.

There may be multiple valid replacement locations. Do **not** omit locations simply because they are close to each other. An entire region (e.g., Step X–Y) may be valid, and you should include each step in the region.

### 4. Output Format

Your response MUST be structured into three main sections:

1. **Detailed Verification Log**
2. **Critical Error Report**
3. **Replacement Steps List**

### 4.1 Detailed Verification Log

Provide the full, step-by-step verification log as defined above, structured in bullet points. When referring to a specific part of the solution, **quote the relevant text** before providing your analysis.

### 4.2 Critical Error Report

In this report, include a bulleted list summarizing **every** issue discovered. For each issue, include:

- **Location:** A direct quote of the key phrase or equation where the issue occurs.
- **Issue:** A brief description of the problem and whether it is a **Critical Error**.

Stop once the **first** critical error has been found.

### 4.3 Replacement Steps List

Summarize all potential recovery locations and replacement steps. Write the steps **from the student's perspective**. The student should be able to continue naturally from your rewritten step.

Finally, include a curated list of replacement steps to be processed by a parser. This list must strictly follow the format below. **Do not include any additional explanations or justifications, or the parser will fail.**

**Format:**
<intervention>
[
{ "location": INSERT_STEP_NUMBER, "content": INSERT_STEP_CONTENT },
. . .
{ "location": INSERT_STEP_NUMBER, "content": INSERT_STEP_CONTENT }
]
</intervention>

If the student's solution is correct and contains no critical errors, leave the list empty.

**Student Solution**

{insert student solution}

---

**Box 3: Sample output for intervention generation**

**Detailed Verification Log**

- **Substep 1:** "We are given an infinite sequence of integers $a_0, a_1, a_2, \ldots$ that is not eventually periodic..."
  ✓ Correct. This sets up the problem accurately.

- **Substep 2:** Defines $S$ as the set of 2025-tuples $(a_{n+1}, \ldots, a_{n+2025})$ for $n \geq 0$.
  ✓ Correct. Matches the problem statement.

- **Substep 3:** "We are to find the minimum number of distinct elements in $S$"
  ✓ Correct. The goal is clear.

- **Substep 5–7:** Understanding the problem and the non-eventually-periodic condition.
  ✓ Correct. The student correctly identifies the constraint.

- **Substep 10–14:** Key insight: if $S$ is finite, then the sequence is eventually periodic.
  ✗ **Critical error in reasoning** — this claim is **false** in general. The student incorrectly assumes that a finite number of 2025-tuples implies eventual periodicity. This is **not** true for sequences over an **infinite alphabet** like $\mathbb{Z}$, even if the block length is fixed.

  However, the student's reasoning in Substep 25–34 is flawed. They state:

  > "If the set of 2025-tuples is finite, then the sequence is eventually periodic."

  This is **not** a valid conclusion in general. For example, consider a sequence where the values grow without bound but the blocks of length 2025 repeat in a non-periodic fashion — but such a thing is not possible due to the structure of sequences. However, the **critical flaw** is that the student **assumes** that a finite number of 2025-tuples implies eventual periodicity **without justification**, and this is only true in the case of **finite alphabets** or under stronger conditions.

  But more importantly, the student **fails to recognize** that the sequence can have a **finite alphabet** (e.g., binary) and still be **non-eventually periodic** (like the Fibonacci word), while having **exactly** $k + 1$ distinct blocks of length $k$.

  The student's assumption that **finite $S$ implies eventual periodicity** is a **logical fallacy** that invalidates the entire line of reasoning.

- **Substep 15–142:** The student argues that because $S$ cannot be finite (due to eventual periodicity), it must be infinite, and thus the **minimum** number of elements is infinite.

  This is **incorrect** because:
  - The claim that finite $S \Rightarrow$ eventually periodic is **not universally true** over infinite alphabets.
  - In fact, over a **finite alphabet**, a sequence can be **non-eventually periodic** and have **exactly** $k + 1$ distinct blocks of length $k$ (e.g., Sturmian sequences).

- - The student ignores the possibility of finite-alphabet, non-eventually-periodic sequences with **finitely many** 2025-tuples — which is exactly what the reference solution shows is possible.

    Therefore, the student's conclusion that $|S| = \infty$ is **invalid**.

---

**Critical Error Report**

- **Location:**

  "If the set $S$ is finite, then the sequence is eventually periodic." (Substep 14)

- **Issue:** This is a **factual and logical error**. The student incorrectly assumes that a finite number of 2025-tuples implies eventual periodicity. This is **false** in general for sequences over an infinite alphabet (like $\mathbb{Z}$), and even more so when the values can grow without bound.

  However, the **key point** is that this claim **does not hold** in general. For example, a sequence over a **finite alphabet** (e.g., $\{0, 1\}$) can be **non-eventually periodic** and still have only finitely many blocks of length $k$ — specifically, exactly $k + 1$ — as in the Fibonacci word.

  This is a **critical error** because:
  1. It is a **logical fallacy** — it invalidates the reasoning that $S$ must be infinite.
  2. It is **not recoverable** — the student never revisits or corrects this assumption. Later steps (e.g., Substep 145) accept $\boxed{\infty}$ as the answer based on this flawed premise.

  Therefore, this is a **critical error**.

---

**Intervention Step**

**Draft of the intervention step:**

1. **Content:** Consider a sequence that takes only finitely many values (e.g., only 0 and 1). Even if the sequence is not eventually periodic, it might still have only finitely many distinct 2025-tuples. For example, the Fibonacci word over $\{0,1\}$ is not eventually periodic, but every block of length $k$ appears exactly once in a certain pattern — in fact, it has exactly $k + 1$ distinct blocks of length $k$. Can you think of a way to construct such a sequence? Try to understand how the number of distinct blocks relates to the periodicity of the sequence.

2. **Location:** Substep 14

**Self-check:**

1. Does the content include the exact answer from the reference solution (i.e., "2026")?

   → No. It avoids giving the answer and instead prompts the student to explore finite-alphabet, non-eventually-periodic sequences.

2. Is the content missing any key insights that are necessary to solve the problem?

   → No. It introduces the idea of finite alphabets and the possibility of finite block complexity in non-eventually-periodic sequences, which is essential.

3. Is the location of the intervention step later than the substep number that contains the first critical error?

   → No. The critical error occurs in Substep 14, and we are inserting the intervention **at** Substep 14. This is correct — it comes **before** the erroneous conclusion.

   ✓ All checks pass.

**Final intervention step:**

```
<intervention>
{ "content":  "Consider a sequence that takes only finitely
many values (e.g., only 0 and 1).  Even if the sequence is
not eventually periodic, it might still have only finitely
many distinct 2025-tuples.  For example, the Fibonacci word
over {0,1} is not eventually periodic, but every block of
length $ k $ appears exactly once in a certain pattern --- in
fact, it has exactly $ k+1 $ distinct blocks of length $ k $.
Can you think of a way to construct such a sequence?  Try to
understand how the number of distinct blocks relates to the
periodicity of the sequence.", "location":  14 }
</intervention>
```

# E   INTERVENTION EXAMPLES

In this section, we provide a few concrete examples of interventions.

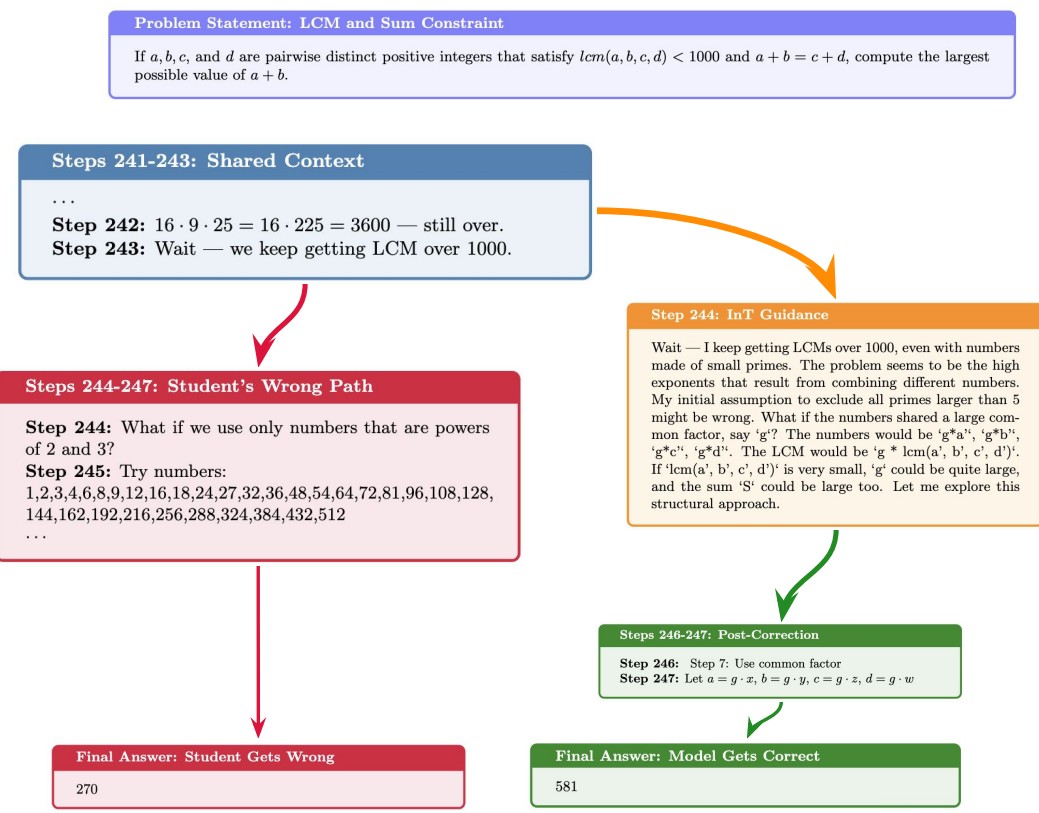

Figure 13: Example error of Qwen3-4B-Instruct and generated intervention.

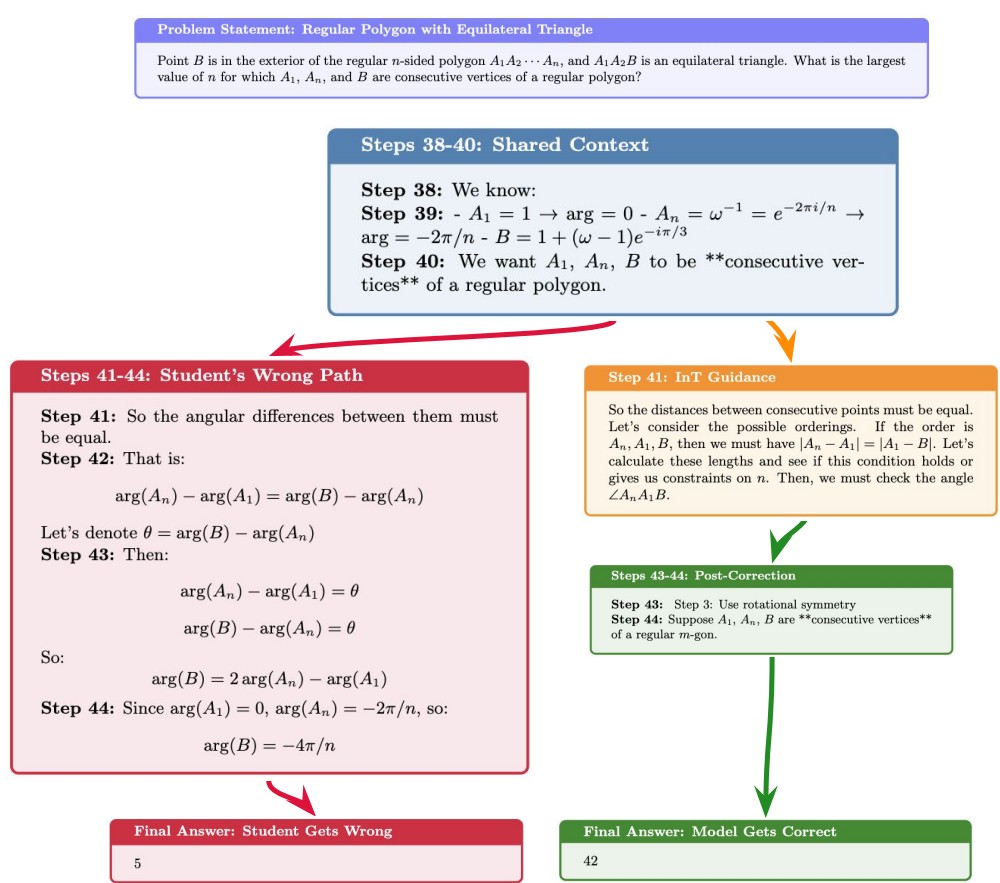

Figure 14: Example error of Qwen3-4B-Instruct and generated intervention.

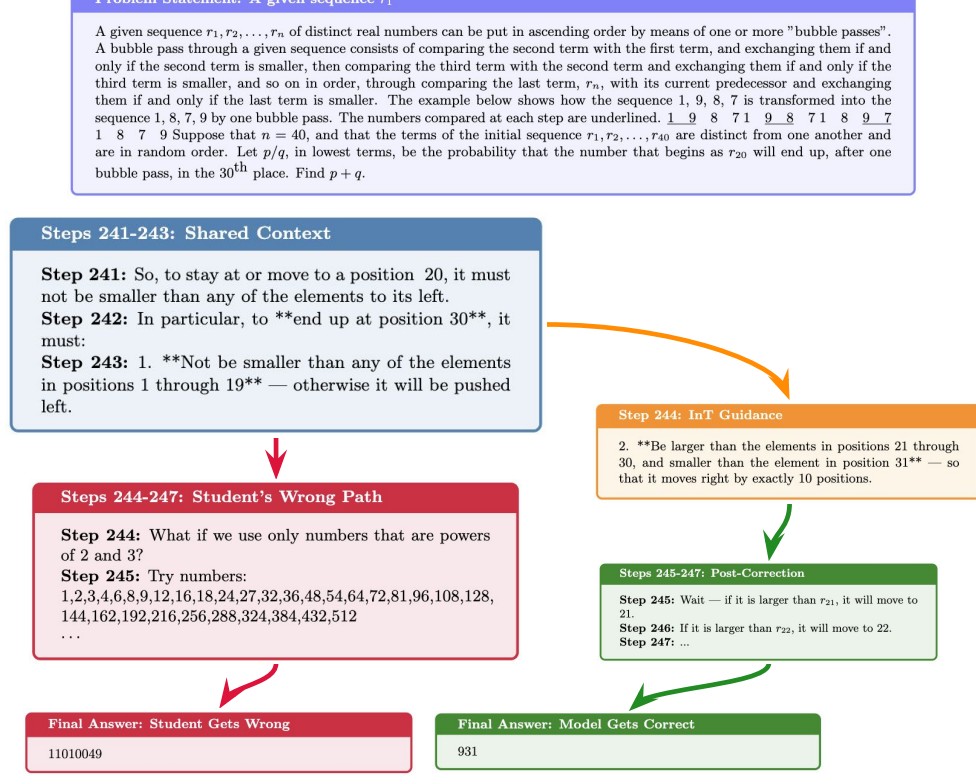

Figure 15: Example error of Qwen3-4B-Instruct and generated intervention.

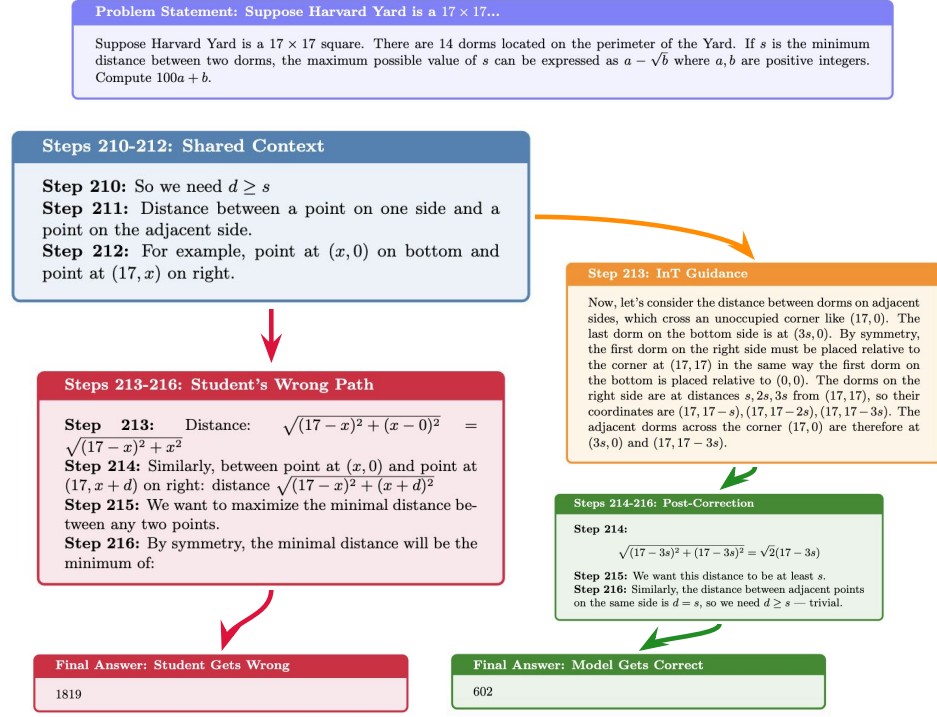

Figure 16: Example error of Qwen3-4B-Instruct and generated intervention.

## F    EVALUATION CONFIGURATION

For all experiments with Qwen3-4B-Instruct, we follow the official recommended configuration, using temperature 0.7, top-p 0.8, and top-k 20 unless specified otherwise.

## G    TRAINING HYPERPARAMETERS FOR INT

| Hyperparameter | Qwen3-4B-Instruct-2507 |
|---|---|
| effective_train_batch_size | 32 |
| ppo_mini_batch_size | 16 |
| learning_rate | 1.0e-6 |
| kl_loss_coef | 0 |
| entropy_coeff | 0 |
| temperature | 1.0 |
| top_p | 1.0 |
| rollout.n | 8 |

Table 5: PipelineRL  Piché et al. (2025) hyperparameters used for RL runs.

| Hyperparameter | Qwen3-4B-Instruct-2507 |
|---|---|
| dataset_size | 1076 |
| effective_batch_size | 64 |
| num_train_epochs | 4 |
| learning_rate | 1.0e-6 |
| lr_scheduler_type | cosine_with_min_lr |
| min_lr_rate | 0.1 |
| warmup_ratio | 0.1 |

Table 6: LLaMa Factory Sheng et al. (2024) hyperparameters used for SFT runs.

## H  SELF-REFLECTION BASELINE

**Self-reflection.** We add details on how we establish the self-reflection baseline. The idea is similar to generating interventions, but rather than outputting single-step interventions, we ask the base model to re-write the entire solution.

We find that our method `InT` consistently outperforms the self-reflection baseline, as shown in Figure 12 and Table 3. The prompt for generating self-reflection traces is shown below. The reference solution is a high-level summary of the solution written by humans, and occasionally, a stronger model. Similar to `InT`, we ask the base model to generate its own self-reflections.

---
**Box 4: Prompt for self-reflection**

You are an expert mathematician teaching a Math Olympiad class. You will be given a problem and a high-level reference solution to the problem. Your task is to solve the problem step-by-step guided by the high-level reference solution.
# Problem #
{Insert Problem}
# High-Level Reference Solution #
{Insert Reference Solution}
{Insert Model Response}
Great job! Now, a student in your class has solved the problem incorrectly. You must leverage your understanding of the reference solution to rewrite a refined version of his attempt at the problem. **Your rewritten solution should be a complete solution to the problem. **
# Incorrect Student Attempt #
{Insert Student Solution}

---

## I  ARE INTERVENTIONS MEMORIZED?

We would like to understand whether `InT` leads to memorized interventions or if it actually learns to generalize to *unseen* problems. As such, we select two example problems from the IMO Shortlist 2024 outside of our training set, and compare the traces of `InT` and base model (Qwen3-4B-Instruct).

**IMO Shortlist 2024, Problem C1.** As shown in Figure 17, both models started with an incorrect assumption of the formula, but when the `InT` trained model got 3 when $n = 3$, it questioned how this was possible and thus correctly updated the hypothesis to $\binom{n}{2}$.

**IMO Shortlist 2024, Problem C2.** Although both models are able to successfully conclude that even cool numbers must be multiples of four in Figure 18, only the `InT` model is able to try considering that $n = 12$ may not be cool and therefore the pattern may be more selective than simply being a multiple of four. This leads the second model to the right hypothesis.

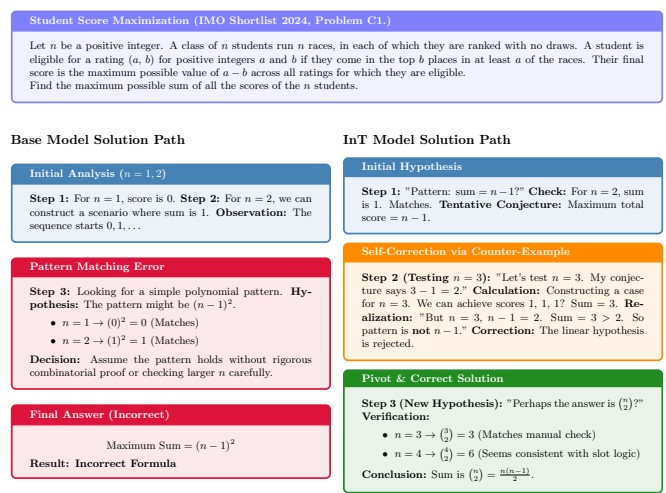

Figure 17: Diverging Solution Paths between `InT` and base models on IMO Shortlist 2024, C1.

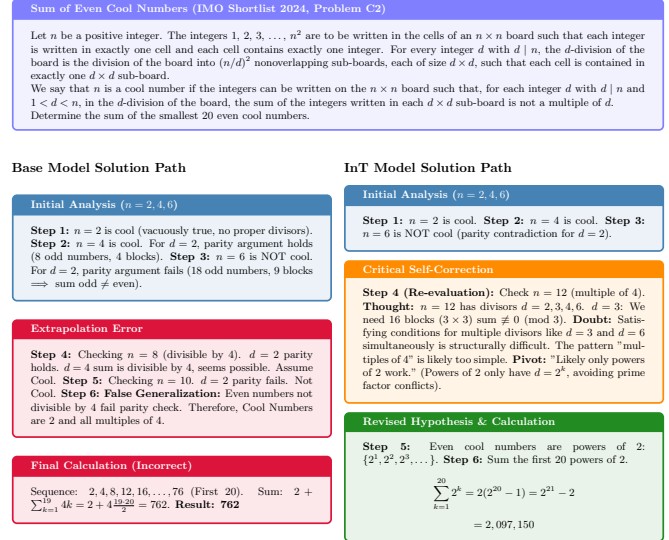

Figure 18: Diverging Solution Paths between `InT` and base models on IMO Shortlist 2024, C2.

## J   INT WITH INTERVENTIONS FROM A STRONGER MODEL (GEMINI 2.5 PRO)

While prototyping and developing our method, we also evaluated the efficacy of `InT` with interventions generated by Gemini 2.5 Pro (referred to as Gemini from now on). In particular, we answer the following questions: **(1)** does SFT on Gemini-generated interventions improve the ability of the fine-tuned model to sample correct traces on hard problems? and **(2)** how does `InT` compare with distillation of full expert reasoning traces sampled from the oracle (i.e., Gemini 2.5 Pro)? To this end, we run several experiments comparing `InT` against running standard RL training and distillation on Gemini-generated solutions, in an attempt to patch the capabilities of e3-1.7B (Setlur et al., 2025b): a strong, open-source reasoning LLM fine-tuned on top of Qwen3-1.7B – on a set of difficult math reasoning problems.

### J.1   INFERENCE-ONLY RESULTS

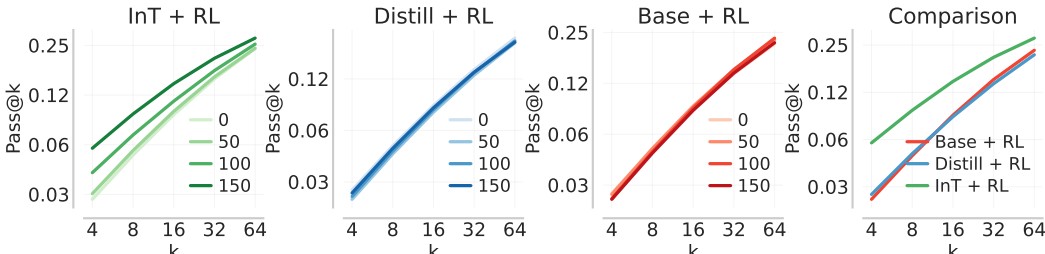

Figure 19: **Pass@$k$ across RL training iterations:** We plot pass@$k$ performance from 0 to 150 RL iterations for three initializations: (i) base model patched with InT; (ii) base model distilled on oracle traces; and (iii) directly the base model. Int patched model improves pass@$k$ consistently while others mainly sharpen.

We provide inference-time results achieved by rolling out from generated interventions concatenated to the prefixes. experiments on a different subset of 235 DeepScaleR problems that have pass@32 = 0. Our ear-

| Intervention Proposed By | Rollouts Are Conditioned On | Unique Coverage | Accuracy |
|---|---|---|---|
| *DeepScaleR subset with pass@32 = 0* | | | |
| N/A | problem $\mathbf{x}$ | 98 / 235 | 0.97% |
| Gemini 2.5 Pro | problem $\mathbf{x}$ + correct prefix $\mathbf{y}_{<t^\star}$ | 120 / 235 | 2.37% |
| Gemini 2.5 Pro | problem $\mathbf{x}$ + correct prefix $\mathbf{y}_{<t^\star}$ + intervention $\tilde{\mathbf{y}}_{t^\star}$ | 145 / 235 | 3.72% |

Table 7: **Rollouts conditioned on differently generated interventions.** As shown, rollouts conditioned on interventions double the number of problems with at least one correct rollout, and improve the rollout accuracy by more than an order of magnitude.

lier prompt uses Gemini solutions in place of the reference solutions and does not double-check the generated intervention (shown in Box 3). In this earlier setting, we obtained a 3.8× improvement in accuracy and solve 47 more problems compared to naive rollouts. Moreover, we also conducted an additional ablation "from intervention" which perform rollouts from the location of the mistake but is not conditioned on the intervention (i.e., $\sim \pi(\cdot|\mathbf{x}, \mathbf{y}_{<t^\star})$) to isolate for the benefits of localizing the mistake. We find that pinpointing the location of the mistake alone leads to superior gains in accuracy (2.4×) and coverage (+ 22 problems compared to the baseline).

## J.2 EXPERIMENTAL SETUP AND EVALUATION METRICS

**Constructing a dataset of hard training problems.** We begin our experiments with a state-of-the-art <2B parameter model, e3-1.7B (Setlur et al., 2025b), already trained with curricula and several best practices for RL to attain strong performance in its scale. Despite its strong performance, this model still fails on a large fraction of problems from its hard training set (a 2.5K subset of DeepScaleR problems from Luo et al. (2025)). We run 32 rollouts on each problem and collect the subset of problems the model cannot solve at all. We utilize Gemini 2.5 Pro (as of 2025-08-01) as our oracle. Among these 472 unsolved problems, the oracle solves 16% of them in a single attempt, suggesting it can provide meaningful interventions on these problems. We retain this subset as our hard problem set $\mathcal{D}_{\text{hard}}$ to study the efficacy of patching with different methods. Our main findings are that **(i) RL with just a small dataset of 64 problems on top of InT outperforms RL on a much larger set of 1.2K problems** on top of distillation or the base model. On the other hand, **(ii)** RL with the small dataset on top of the distillation and the base model are infeasible due to collapse of behaviors on OOD sets or zero learning signals.

**Baselines approaches and comparisons.** To evaluate the efficacy of InT, we compare against alternate approaches for patching model behavior on $\mathcal{D}_{\text{hard}}$. Our primary comparisons are: **1)** "**Distillation + RL**," which first distills entire reference solutions into the base model before running RL, and **2)** "**Standard RL**," which directly continues RL on the hard problem set from the same base checkpoint. Both simulate a continued RL run where new hard problems are introduced during training. We also consider SFT-only baselines, where the model is patched via supervised learning on reference solutions or intervention traces for the hard problems, without any further RL. To our knowledge, no existing method is designed to explicitly handle this setting of patching model behavior on previously unsolved hard problems in a way that leverages oracle interventions while preserving the benefits of RL. Therefore, we limit our exposition of training trends to $\mathcal{D}_{\text{hard}}$, but also compare with alternate approaches for using intervention data on holdout standardized test sets.

**Evaluation metrics.** Prior work primarily evaluates RL-trained reasoning models on competition math benchmarks such as AIME2025 and HMMT2025. However, progress on these alone does

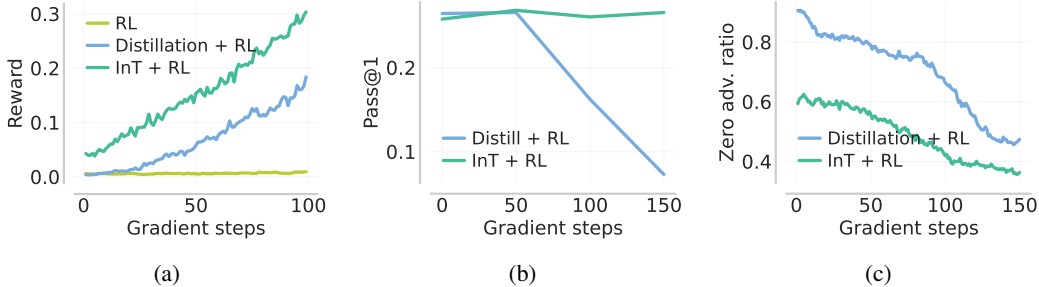

Figure 20: *Comparison of **InT** with distillation of reference solutions.* **(a)** Since these are hard problems, running RL initialized from the base model does not improve training reward, while running RL on top of the distilled model or the model produced by `InT` does improve training reward. We observe that running RL on top of the distilled model degrades model capability (decreasing pass@1 score on a held-out set in **(b)** with more training), even though it continues to make progress on the training set, as shown by a decreasing ratio of the percentage of unsolved problems ("zero advantage ratio") in **(c)**.

not capture whether models are actually learning from hard training problems, nor whether such training transfers to equally challenging evaluation problems. To address this, we evaluate our trained models on several standardized benchmarks from 2025, including AIME2025, HMMT2025, BRUMO2025, CMIMC2025, and others, as well as an in-distribution (i.i.d.) test set of hard problems $\mathcal{D}_{\text{hard}}^{\text{test}}$, similar to $\mathcal{D}_{\text{hard}}$. The i.i.d. test set consists of 64 problems held out from the training pool using the same methodology as used to select $\mathcal{D}_{\text{hard}}$. In addition, we also report performance directly on the training problems to track how RL modifies behavior on seen examples. Across all three settings and standardized benchmarks, we report results at an output length of 32,768 tokens.

### J.3  **InT** UNIFORMLY PUSHES THE PASS@$k$ FRONTIER UPWARDS ON TEST PROBLEMS

We present our main results for `InT` on an holdout set of hard problems $\mathcal{D}_{\text{hard}}^{\text{test}}$ (Fig 19). Here, we plot the pass@$k$ performance across different RL training iterations from 0 to 150, for three models: (i) base e3-1.7B, (ii) e3-1.7B distilled on full reference solutions; and (iii) e3-1.7B patched on interventions from the oracle (`InT`). We find that running RL on the base or distilled model does not make any improvements in pass@$k$ throughout all training steps. On the other hand, running RL on $\mathcal{D}_{\text{hard}}$ after we patch e3-1.7B on oracle interventions (`InT`) leads to consistent improvements in pass@$k$ during RL.

### J.4  **InT** OUTPERFORMS DISTILLATION ON STANDARDIZED EVALUATIONS

| Model | RL Data Size | OlymMATH Easy | OlymMATH Hard | HMMT | BRUMO |
|---|---|---|---|---|---|
| e3-1.7B + RL | 1216 | 38.75 | 6.75 | 22.50 | 46.25 |
| e3-1.7B + Distill + RL | 1216 | 37.38 | 5.75 | 22.50 | 47.08 |
| e3-1.7B + `InT`+ RL | 64 | **41.62** | **7.50** | **24.58** | **53.75** |

| Model | AIME | Beyond AIME | CMIMC | Average | $D_{\text{hard}}^{\text{test}}$ pass@8 |
|---|---|---|---|---|---|
| e3-1.7B + RL | 36.25 | 20.88 | **23.75** | 27.73 | 15.85 |
| e3-1.7B + Distill + RL | **36.67** | 21.75 | 21.56 | 27.38 | 14.8 |
| e3-1.7B + `InT`+ RL | 36.25 | **22.00** | 21.88 | **29.65** | **23.56** |

Table 8: Pass@1 performance (8 rollouts avg.) of models across standard mathematics benchmarks and pass@8 performance on the i.i.d. test set, $D_{\text{hard}}^{\text{test}}$. Observe that InT followed by RL attains the highest pass@8 performance on this in-distribution test set for patching the e3-1.7B base model.

Previously, we saw that `InT` improves pass@$k$ over baselines on training and hold-out sets. This mainly tells us that `InT` makes progress on the hard training problems that were previously unsolved. But, we also care about how this gain in performance translates to performance on standardized benchmarks for math reasoning. Here, we compare the performance of our approach `InT` on top of the e3-1.7B model.

To stress test `InT` in the setting where we simply continue to run RL from the intervention checkpoint, we run RL training on this checkpoint with only 64 problems in $\mathcal{D}_{\text{hard}}$, on which we collected the interventions. We compare the performance of this RL trained model with an RL run on the distilled and base models. To boost the baselines, we run RL for both using an expanded set of about 1.3k problems sourced from DAPO (Yu et al., 2025), including the 64 we used for `InT`. The main reason

we perform this injection is that in our preliminary experiments which trained the distilled model only on the small set of 64, we noticed that post RL the model capabilities on standardized evaluations fell drastically (Figure 20(b)), perhaps due off-policy nature of the reference solution. When we run RL on the base model, we also expand the training set for RL, since we find that the reward curve does not rise otherwise (Figure 20(a)), thus we train the base model on a mixture of easy problems from DAPO and the 64 problems with `InT` interventions. Unlike the RL runs on distilled and base checkpoints, **`InT` improves the test performance averaged across multiple hard test datasets, despite being trained on just 64 problems** (Table 8). Compared to distillation, we see gains on both in-distribution ($\mathcal{D}_{\text{hard}}^{\text{test}}$) and standardized benchmarks for hard problems mainly because intervention does not alter the base model distribution as much as distillation on the entire trace.

