# OpenReview forum: "InT: Self-Proposed Interventions Enable Credit Assignment in LLM Reasoning"
_ICLR.cc/2026/Conference — ICLR 2026 Poster_

### Official Review · Reviewer_MBkX · 2025-10-28

**Soundness:** 3
**Presentation:** 2
**Contribution:** 3
**Rating:** 4
**Confidence:** 5

**Summary:**

This paper introduces Interventional Training (InT), a method to improve reinforcement learning (RL) training of LLMs on hard reasoning problems where the model consistently fails to generate correct rollouts (i.e., receives zero rewards). The key idea is to perform targeted credit assignment by identifying and correcting a single intermediate reasoning step where the model goes wrong, using a lightweight oracle (e.g., a stronger LLM). The corrected step is then used to patch the model via supervised fine-tuning (SFT), enabling it to generate counterfactual correct traces and resume RL training.

Key contributions:
1. A method to collect single-step interventions from an oracle to correct reasoning errors;
2. Fine-tune the model on its own prefix + oracle correction, avoiding full trace cloning;
3. Resume RL from the patched model, now capable of receiving non-zero rewards on previously unsolvable problems;
4. InT solves 14–16% more hard problems and improves pass@k on both in-distribution and standardized math benchmarks.

**Strengths:**

S1: This paper addresses a real and underexplored problem—RL stalling on zero-reward problems due to execution errors in long reasoning traces.

S2: InT is lightweight, requiring only ~100 short interventions, and avoids the cost of full oracle traces or dense reward models.

S3: This paper demonstrates consistent improvements over baselines (e.g., distillation, standard RL) on both training and test sets across multiple benchmarks.

S4: Unlike full trace distillation, InT retains model diversity and avoids catastrophic forgetting by only patching on-policy prefixes.

S5: This paper connects to credit assignment and on-policy vs. off-policy learning, with clear ablations and training dynamics analysis.

**Weaknesses:**

W1: This work requires access to a stronger oracle model (e.g., Gemini 2.5 Pro), which may not be available or practical in many settings. The paper does not explore weaker or human oracles or automated error detection.

W2: How to define a step and how to determine whether the split step is reasonable? What is the maximum number of steps after segmentation in this work?

W3: This work was evaluated only on math reasoning tasks with verifiable answers. Generalization to open-ended reasoning, symbolic tasks, or subjective domains is unclear.

W4: RL is only run on 64 hard problems, which raises questions about scalability and stability of the method when applied to larger or more diverse datasets.

W5: While the paper shows where errors occur, it does not analyze what types of errors are fixed, how often the model recovers, or whether it learns to generalize from interventions.

W6: The main baselines are standard RL and full trace distillation, but the paper does not compare with process reward models (PRMs), self-correction, or iterative refinement methods, which are more closely related.

W7: In addition, there are many typos, such as blank pages (page 13, page 17), and incorrect references (Table 3, Table 4, Table5)

**Questions:**

Q1: How does InT perform with weaker or human oracles? Can the method still work if the oracle is not significantly stronger than the base model?

Q2: What happens if the oracle misidentifies the error or provides a suboptimal fix? Is there a mechanism to filter or validate interventions?

Q3: Does the model learn to self-correct, or does it just memorize interventions? How does it behave on similar but unseen problems?

Q4: Can InT be scaled to larger problem sets or more diverse domains? What are the computational or stability limits of continued RL after patching?

Q5: How does InT compare to process reward models (PRMs) or iterative self-refinement? Would training a PRM be more efficient or effective in the long run?

Q6: Why not train the model to predict its own interventions? Could InT be extended to automate the oracle role via meta-learning or self-critique?

---

> ### Author Response · Authors · 2025-11-24
> **Rebuttal by the Authors (Part 1/4)**
>
> Thank you for the review!  Since the original submission, we have made substantial updates to the work. Most importantly, we have
>
> 1. Added several new experiments that **remove entirely the dependence on an external oracle model (Gemini 2.5 Pro).** The base model now generates its own interventions, which clarifies the core contributions of InT: it now becomes clear that InT is not simply distilling responses from a stronger teacher. Instead, InT benefits from external signals (gold solutions) to produce effective interventions, highlighting it as a self-contained training strategy. We will include these new experiments as the main results of the paper.
>
> 2. **Introduced a new critique-based self-refinement baseline as requested,** where the base model revises its entire solution, rather than producing a localized single-step intervention. A full description and the exact prompt used for self-refinement are provided in **Appendix J**.
>
> 3. **Evaluated models on more challenging, standardized benchmarks:** IMO-Bench [1], AMO-Bench [2], and Apex Shortlist [3], and the newly released, uncontaminated HMMT November 2025 problems, which we scraped from [4]. We run RL for *400 steps* and observe that training with InT leads to a near 10% improvement in average performance compared to the base model, and 3% improvement compared to the strongest baseline.
>
> Below is a table summarizing our final results using **Qwen3-4B-Instruct** as the base model:
>
> | Model    | IMO-Bench | HMMT 2025 Nov | AMO-Bench pass@8 | Apex Shortlist pass@8 | Average |
> |------------------------|-----------|---------------|-------------------|------------------------|---------|
> | Base | 11.68 | 41.61 | 26.24 | 20.79 | 25.08 |
> | + RL                | 23.46     | 46.46         | 35.21             | 22.72                  | 31.96   |
> | + Gold SFT + RL          | 11.56     | 27.45         | 25.19             | 20.51                  | 21.18   |
> | + Self-refine SFT + RL   | 15.53     | 38.65         | **36.72**         | 23.93                  | 28.71   |
> | **+ InT + RL (Ours)**     | **25.62** | **49.77**     | 36.16             | **28.22**              | **34.94**   |
>
> We also show the **pass@k curves** across training on the test set **in the figure below** to evaluate if InT is meaningfully able to go beyond the capabilities of the base model or simply “sharpens” what the base model already knows. Across both sets of results, **InT achieves the strongest overall performance and a higher pass@k value for all $k$ values we could evaluate**, indicating that patching the LLM with InT indeed improves its overall capabilities.
>
> **Figure:** [Pass@k curves over training](https://ibb.co/pvWfKqH3)
>
> If these revisions address your concerns, we kindly request that you consider updating your score. Thank you!
>
> [1] [IMO-Bench](https://imobench.github.io/)
>
> [2] [AMO-Bench](https://amo-bench.github.io/)
>
> [3] [Apex Shortlist](https://huggingface.co/datasets/MathArena/apex-shortlist)
>
> [4] [HMMT Website](https://www.hmmt.org/www/archive/problems)

---

> ### Author Response · Authors · 2025-11-24
> **Rebuttal by the Authors (Part 2/4)**
>
> We answer your questions in more detail below.
>
> > **W1: This work requires access to a stronger oracle model (e.g., Gemini 2.5 Pro), which may not be available or practical in many settings. The paper does not explore weaker or human oracles or automated error detection.**
>
> > **Q1: How does InT perform with weaker or human oracles? Can the method still work if the oracle is not significantly stronger than the base model?**
>
> > **Q6: Why not train the model to predict its own interventions? Could InT be extended to automate the oracle role via meta-learning or self-critique?**
>
>
> We have addressed this concern in our updated experiments which generates interventions **without any dependence on external oracles (Gemini 2.5 Pro)**. Instead, the **base model now generates its own interventions**, relying only on an openly available, open-source model. This removes the cost and availability constraints of proprietary oracles and opens up the method to practitioners who do not have access to them.
>
> A brief summary of our main results is provided above, with further details in **Appendix H**. Taken together with the results already present in the paper, these new results highlight that InT does not need to rely on oracles for interventions. Perhaps because verifying an existing solution against a gold solution is easier than generating a solution from scratch, we find that using base models as the oracle is sufficient for generating interventions.
>
> > **W2: How to define a step and how to determine whether the split step is reasonable? What is the maximum number of steps after segmentation in this work?**
>
> We define a *step* as a contiguous block of text separated by a double newline (“\n\n”). This segmentation scheme is consistent with the output format of the Qwen model family, on which our work is built. In general though, this definition is flexible and can change depending upon the model. For example, [1, 2] define step as a block separated by keywords such as “Wait”, “alternatively”, “maybe”, etc. and this definition is also possible, and we would not expect it to affect the efficacy of our approach. However, since we were using a Qwen model, we chose to stick to the output format prescribed by this model family.
>
> In these models, each block separated by “\n\n” is treated as a self-contained reasoning unit, so segmenting by “\n\n” yields steps that reliably align with coherent chains of thought. For clarity, we include an illustrative example below:
>
> ```
> Okay, so I need to find the smallest positive perfect cube that can be written as the sum of three consecutive integers. Hmm, let me think about how to approach this.
>
>
> First, let me recall that three consecutive integers can be represented as n, n+1, and n+2. Their sum would be n + (n+1) + (n+2). Let me calculate that:
>
>
> n + (n + 1) + (n + 2) = 3n + 3. So, the sum is 3n + 3, which simplifies to 3(n + 1). Therefore, the sum of three consecutive integers is three times (n + 1).
>
> So, the problem is asking for the smallest positive perfect cube that can be expressed as 3(n + 1). Therefore, we need 3(n + 1) to be a perfect cube. So, 3(n + 1) = k³, where k is a positive integer. Therefore, n + 1 = k³ / 3. Since n has to be a positive integer, k³ must be divisible by 3. Therefore, k must be divisible by 3.
>
> …
> ```
>
> **We do not impose an explicit upper bound on the number of steps.** Instead, we generate model rollouts up to a 16K-token limit and segment them afterward. Within this limit, we find that traces typically contain up to roughly 400 steps. Figure 4 in the paper, which reports the distribution of intervention step indices, further illustrates this range and may be helpful for interpreting the segmentation behavior.
>
> [1] [Beyond the 80/20 Rule: High-Entropy Minority Tokens Drive Effective Reinforcement Learning for LLM Reasoning](https://arxiv.org/abs/2506.01939)
>
> [2] [Optimizing Test-Time Compute via Meta Reinforcement Fine-Tuning](https://arxiv.org/abs/2503.07572)

---

> ### Author Response · Authors · 2025-11-24
> **Rebuttal by the Authors (Part 3/4)**
>
> > **W3: This work was evaluated only on math reasoning tasks with verifiable answers. Generalization to open-ended reasoning, symbolic tasks, or subjective domains is unclear.**
>
> To broaden our evaluation scope, we have added **substantially harder benchmarks** - IMO Bench, AMO Bench, and the Apex Shortlist. Please refer to the table above for the full results (details in Appendix H).
>
> **Benchmark Difficulty.** On top of mid-level olympiad sets (AIME, HMMT, OlympiadBench, etc.), which are increasingly saturated for current models, we add **new benchmarks of significantly higher difficulty:**
>
> - IMO Bench: IMO-level problems curated by IMO medalists.
>
> - AMO Bench: IMO-level Olympiad problems.
>
> - Apex Shortlist: problems where even top frontier models achieve ~50% accuracy.
>
> **Beyond math reasoning.** Generalizing InT to open-ended reasoning (e.g., IMO Proof Bench), symbolic tasks, and subjective domains is a natural next step, and we plan to explore these directions in future work. We have added this as a limitation in the paper, and a direction for future work. Would you have recommendations for any benchmarks or domains to study for this purpose?
>
> > **W4: RL is only run on 64 hard problems, which raises questions about scalability and stability of the method when applied to larger or more diverse datasets.**
>
> > **Q4: Can InT be scaled to larger problem sets or more diverse domains? What are the computational or stability limits of continued RL after patching?**
>
> We have **substantially expanded the training set** in our latest experiments from 64 to **1000 problems**. The method remained stable at this scale and yielded strong improvements on highly challenging benchmarks, as shown in the table above. Please refer to Appendix H for detailed results.
>
> That said, prior work in our submission has shown that running InT with very few problems is also effective and also faster for iteration speed during the course of research, which is the reason why we chose to do so during the submission. We did believe though that results from training on 64 problems should transfer to the larger dataset setting. Indeed that is the case, as proven by our new results with 1000 problems that we have now added to Appendix H.
>
> > **W5: While the paper shows where errors occur, it does not analyze what types of errors are fixed, how often the model recovers, or whether it learns to generalize from interventions.**
>
> > **Q3: Does the model learn to self-correct, or does it just memorize interventions? How does it behave on similar but unseen problems?**
>
> We do analyze the types of errors that are fixed. As shown in **Figure 4**, InT corrects both high-level reasoning errors and lower-level execution mistakes, and we report the distribution of where these errors occur. **Figure 2** and **Appendix B** provide concrete examples of such corrections.
>
> **Figure:** [Examples of models trained with InT correcting mistakes on unseen problems](https://ibb.co/p9h3ZYZ)
>
> In our revision, we have also added additional examples in the figure above, illustrating how InT fixes specific mistakes on *unseen* problems, providing further evidence that the model learns to generalize from interventions.

---

> ### Author Response · Authors · 2025-11-24
> **Rebuttal by the Authors (Part 4/4)**
>
> > **Q2: What happens if the oracle misidentifies the error or provides a suboptimal fix? Is there a mechanism to filter or validate interventions?**
>
> As indicated in **Section 4.2**, an intervention is only kept if conditioning the model on the patched prefix produces at least one correct rollout. If the fix does not actually repair the reasoning, it is discarded and never enters the SFT data. This ensures that only validated, outcome-correct interventions are learned. As shown in Table 1, rollouts conditioned on interventions increase the number of unique problems for which a correct trace is discovered compared to naïve rollouts, demonstrating the effectiveness of interventions.
>
> In our newest experiments shown in the table below, we observe the same pattern at a larger scale using interventions self-generated by the base model. After producing interventions for 4.5k difficult problems, we perform 32 rollouts per problem conditioned on these interventions. On a subset of 334 problems, we compare naïve rollouts to intervention-conditioned rollouts and find that **conditioning on interventions doubles the number of problems with at least one correct rollout** and **improves accuracy by over an order of magnitude**, further reinforcing that interventions substantially enhance the model’s ability to recover correct reasoning traces.
>
> | Configuration     | Nonzero Acc. | Accuracy   |
> |-------------------|--------------|------------|
> | Naive             | 40/334       | 0.0984%    |
> | With intervention | 80/334       | 1.56%      |
>
>
> > **W6: The main baselines are standard RL and full trace distillation, but the paper does not compare with process reward models (PRMs), self-correction, or iterative refinement methods, which are more closely related.**
>
> > **Q5: How does InT compare to process reward models (PRMs) or iterative self-refinement? Would training a PRM be more efficient or effective in the long run?**
>
> To address this concern, we discuss two closely related methods:
>
> 1. **Self-refinement baseline.** We implemented a self-refinement method where the model critiques its full reasoning chain and revises its answer, making it directly comparable to InT (implementation details in **Appendix J**). As shown the table above (**more details in Appendix H**), InT consistently outperforms self-refinement, perhaps indicating that **targeted first-error interventions provide clearer credit assignment than global critiques**, which may dilute the learning signal.
>
> 2. **PRMs.** PRMs are relevant, but high-quality PRMs are not yet available for the newer base models we use and prior work has shown that PRMs underperform pure RL baselines, especially on difficult problems [1], and training a PRM from scratch requires significant additional work. We can include experiments using an available open-source option (e.g., Qwen2.5-Math-PRM-7B) if the reviewer feels this comparison is important, and would welcome guidance on the setup they consider most appropriate, and we will be happy to add it in the remainder of this rebuttal period.
>
> [1] [​​Is PRM Necessary? Problem-Solving RL Implicitly Induces PRM Capability in LLMs](https://arxiv.org/abs/2505.11227)
>
> > **W7: In addition, there are many typos, such as blank pages (page 13, page 17), and incorrect references (Table 3, Table 4, Table5)**
>
> Thank you for noting these issues. We have corrected the typos, removed the blank pages (p.13, p.17), and fixed the incorrect references (Table 3, Table 4, Table 5) in the updated manuscript.

---

> > ### Author Response · Authors · 2025-11-26
> > **Did our Rebuttal Address your Concerns?**
> >
> > Dear Reviewer,
> >
> > Since there is only 1 week left in the rebuttal period, it would be helpful for us to know if our rebuttal above addressed all your questions and concerns or if any others still remained? We are happy to discuss further and provide more evidence if it is helpful. If all your concerns are addressed we would be grateful if you could acknowledge that soon.
> >
> > Thanks,
> >
> > Authors

---

> > > ### Author Response · Authors · 2025-11-27
> > >
> > > Dear Reviewer,
> > >
> > > We apologize for pinging you again!
> > >
> > > Given that there are only a four more days left in the rebuttal period, we wanted to reach out once more to see if our rebuttal above addressed all your concerns or if any others still remained? We are happy to discuss further and provide more evidence in the few days that are left. If all your concerns are addressed we would be really grateful if you could acknowledge that soon.
> > >
> > > Thanks,
> > >
> > > Authors

---

### Official Review · Reviewer_ikPh · 2025-10-28

**Soundness:** 4
**Presentation:** 4
**Contribution:** 3
**Rating:** 6
**Confidence:** 2

**Summary:**

This work is based on the observation that RL is difficult to train and converge when the initial success rate is very low. To address this issue, the paper proposes an Interventional Training (InT) framework that replaces challenging reasoning steps with oracle interventions. The base model is then tuned using these oracle interventions. Experimental results demonstrate the effectiveness of this approach.

**Strengths:**

1. The paper is well organized and clearly written.
2. The motivation is well-defined. The authors also discuss other approaches they have tried and provide reasonable explanations for their limitations, which makes the proposed framework convincing.
3.  Experimental results support their findings.

**Weaknesses:**

1. Literature review is not sufficient. There is extensive prior work on improving RL performance using oracle guidance or hints. However, this paper does not adequately discuss or compare with such related works. For example, recent works have explored combining RL and SFT to enhance RL performance.
2. Although the authors repeatedly mention credit assignment, the explicit connection between InT framework and credit assignment is unclear.
4. Experimental validations are not sufficient. The experiments are conducted with only one base model and one oracle model. It would be helpful to include results across multiple model pairs.

**Questions:**

1. How do the authors identify where oracle interventions should be pinpointed? How to control the length of each generated intervention?

2. Why does the distilled model perform worse than the base model? Does this suggest that the teacher model lacks sufficient reasoning capability?

---

> ### Author Response · Authors · 2025-11-24
> **Rebuttal by the Authors (Part 1/3)**
>
> Thank you for the review! We have updated our work significantly since the submission. Most notably, we have:
>
> 1. **As requested, introduced an additional (and significantly weaker) oracle, the base model itself (Qwen3-4B-Instruct), to generate interventions.** Being able to use the base model to self-generate its own interventions strongly verifies the experimental validity of InT: it now becomes clear that InT is not simply distilling responses from a stronger teacher. Instead, InT benefits from external signals (gold solutions) to produce effective interventions, highlighting it as a self-contained training strategy. We will include these new experiments as the main results of the paper.
>
> 2. **Evaluated models on more challenging, standardized benchmarks:** IMO-Bench [1], AMO-Bench [2], and Apex Shortlist [3], and the newly released, uncontaminated HMMT November 2025 problems, which we scraped from [4]. We run RL for *400 steps* and observe that training with InT leads to a near 10% improvement in average performance compared to the base model, and 3% improvement compared to the strongest baseline.
>
> Below is a table with our final results using **Qwen3-4B-Instruct** as the base model:
>
> | Model    | IMO-Bench | HMMT 2025 Nov | AMO-Bench pass@8 | Apex Shortlist pass@8 | Average |
> |------------------------|-----------|---------------|-------------------|------------------------|---------|
> | Base | 11.68 | 41.61 | 26.24 | 20.79 | 25.08 |
> | + RL                | 23.46     | 46.46         | 35.21             | 22.72                  | 31.96   |
> | + Gold SFT + RL          | 11.56     | 27.45         | 25.19             | 20.51                  | 21.18   |
> | + Self-refine SFT + RL   | 15.53     | 38.65         | **36.72**         | 23.93                  | 28.71   |
> | **+ InT + RL (Ours)**     | **25.62** | **49.77**     | 36.16             | **28.22**              | **34.94**   |
>
>
> We also show the **pass@k curves** across training on the test set **in the figure below** to evaluate if InT is meaningfully able to go beyond the capabilities of the base model or simply “sharpens” what the base model already knows. Across both sets of results, **InT achieves the strongest overall performance and a higher pass@k value for all $k$ values we could evaluate**, indicating that patching the LLM with InT indeed improves its overall capabilities.
>
> **Figure:** [Pass@k curves over training](https://ibb.co/pvWfKqH3)
>
> Note: in the results above, **self-refinement** refers to a new baseline where the base model revises its entire solution rather than producing a localized single-step intervention. This baseline is important because it isolates the value of *targeted* interventions: if simple whole-solution rewriting were sufficient, self-refinement should match or exceed InT. Instead, our results show that localized interventions provide a much stronger training signal by focusing the model on the specific point of failure in its reasoning. A full description and the exact prompt used for self-refinement are provided in **Appendix J**.
>
> If these updates address your concerns, we would be very grateful if you could raise your score. Thank you!
>
> [1] [IMO-Bench](https://imobench.github.io/)
>
> [2] [AMO-Bench](https://amo-bench.github.io/)
>
> [3] [Apex Shortlist](https://huggingface.co/datasets/MathArena/apex-shortlist)
>
> [4] [HMMT Website](https://www.hmmt.org/www/archive/problems)

---

> ### Author Response · Authors · 2025-11-24
> **Rebuttal by the Authors (Part 2/3)**
>
> We answer your questions in more detail below.
>
> > **Experimental validations are not sufficient. The experiments are conducted with only one base model and one oracle model. It would be helpful to include results across multiple model pairs.**
>
> We would like to first clarify that in our existing work, experiments are done with two base models, e3-1.7B and Qwen3-4B-Instruct, as shown in section 6. Thus, the requirement for evaluating **multiple base models** is already satisfied in the current submission.
>
> In addition, over the course of the rebuttal period, we extended our results on the Qwen3-4B-Instruct base model by using a **much weaker oracle model - Qwen3-4B-Instruct itself - to generate its own interventions.** This not only introduces *another* oracle model (thereby satisfying the request for **multiple oracles**) but also represents a **major relaxation** on the assumption of a strong external oracle, completely removing reliance on Gemini 2.5 Pro and demonstrating that InT remains effective even under the base model’s own supervision. We show these results in the table above and include more details in **Appendix H**
>
> > **Why does the distilled model perform worse than the base model? Does this suggest that the teacher model lacks sufficient reasoning capability?**
>
> The distilled model performs worse primarily because the **teacher’s traces are more off-policy** relative to the base model, not because the teacher model is inherently bad. When the training distribution shifts too far from the model’s own policy, it can induce pathologies in optimization, inhibiting the model from improving substantially during RL. This effect has also been documented in concurrent works [1, 2, 3].
>
> **Figure:** [InT facilitates on-policy learning](https://ibb.co/KcYqpRQ4)
>
> Consistent with these findings, we add a new analysis showing that InT **produces the most on-policy traces**, and that SFT on InT traces achieves the strongest performance on both train and test sets as shown **in the figure above**. We provide further details in Appendix I.
>
> [1] [Thinking Machines on-policy distillation blog](https://thinkingmachines.ai/blog/on-policy-distillation/)
>
> [2] [Retaining by Doing: The Role of On-Policy Data in Mitigating Forgetting](https://arxiv.org/abs/2510.18874)
>
> [3] [The False Promise of Imitating Proprietary LLMs](https://arxiv.org/abs/2305.15717)
>
> > **Literature review is not sufficient. There is extensive prior work on improving RL performance using oracle guidance or hints. However, this paper does not adequately discuss or compare with such related works. For example, recent works have explored combining RL and SFT to enhance RL performance.**
>
> We expand our positioning relative to three contemporaneous approaches that combine RL with oracle guidance or expert-driven hints: QuestA [1], AdaBack [2], BREAD [3]. **Please let us know if we missed some other related work, and we will be happy to add or discuss it.**
>
> **Comparison to Hint-Guided RL Methods.** Prior works combine RL with expert signals, relying on *gold partial-solution prefixes* the model conditions on before rollouts. These methods never modify the model’s own on-policy reasoning when an error occurs. In contrast, **InT directly intervenes** in the model’s trajectory: when a rollout fails, InT pinpoints the first incorrect step and injects a targeted natural-language correction *at that specific point*, then fine-tunes on the corrected prefix before RL continues. This changes the causal structure of the model’s step-by-step reasoning and provides more precise credit assignment than scaffold-only approaches.
>
> **Benchmark Difficulty.** Prior hint-guided RL methods focus on benchmarks such as MATH, GSM8K, NuminaMath-CoT, OpenR1-Math-220K, and mid-level olympiad sets (AIME, HMMT, OlympiadBench, BRUMO25). These are increasingly saturated for current reasoning models. **In our new updates, InT is evaluated on substantially harder tasks,** including IMO-Bench (full IMO problems curated by medalists, from a very recently released benchmark where Claude Opus / Sonnet 4 performs worse than our model), AMO-Bench (IMO-level or harder), and the Apex Shortlist (frontier-level problems with ~50% accuracy for top models). These settings go well beyond the difficulty considered in QuestA, AdaBack, or BREAD.
>
> One additional clarification that we would like to make is that our work compares against several baselines that combine RL and SFT to enhance RL performance, such as SFT on oracle traces directly and SFT on self-refinement traces.
>
> **Finally, we note that our approach is orthogonal to hint conditioning**: we could apply InT, which occurs at the SFT phase, in conjunction with hint-conditioning for guiding rollout generation during RL. This makes our contribution orthogonal to these prior methods discussed.
>
> [1] [QuestA](https://arxiv.org/abs/2507.13266)
>
> [2] [AdaBack](https://arxiv.org/abs/2506.18110)
>
> [3] [BREAD](https://arxiv.org/abs/2506.17211)

---

> > ### Author Response · Authors · 2025-11-24
> > **Rebuttal by the Authors (Part 3/3)**
> >
> > > **Although the authors repeatedly mention credit assignment, the explicit connection between InT framework and credit assignment is unclear.**
> >
> > One of the key requirements for successful RL training is good credit assignment: given a reward, we need some indication of how various actions (in our case of LLM reasoning, this corresponds to the steps or tokens generated) contributed to its attainment. We believe that many difficult problems remain unsolvable by naive RL training because credit assignment is poor: despite receiving zero rewards on the problem, the model cannot identify the exact reasoning steps that contributed to the incorrect final answer. Instead, outcome-reward RL simply trains the model to downweight the entire response, potentially jeopardizing the model from attaining the right final answer ever again, because it is disincentivized from even producing the correct sets of steps that appeared in this trace that were downweighted due to some imperfect steps. This viewpoint is discussed in prior work [1]
> >
> > InT aims to ameliorate this concern by using an oracle (a model with access to the gold solution) to detect and then correct these incorrect steps via injection of interventions (“patching”). By detecting the exact steps at which mistakes occur during training while reinforcing correct steps prior to the mistake, InT attributes zero rewards directly to the detected mistakes by upweighting counterfactual interventions. This allows the training procedure to localize attainment of reward to a particular step, and enables RL to hone in onto this step during training.
> >
> > Although our approach is far from being the only way to do credit assignment, it offers a cheap and scalable solution by simply instructing the base model to propose an intervention given a gold human-written solution, which is widely available in math datasets. In the table above, we show that InT yields superior results when compared to standard RL and other baselines.
> >
> > [1] [RL on Incorrect Synthetic Data Scales the Efficiency of LLM Math Reasoning by Eight-Fold](https://arxiv.org/abs/2406.14532)
> >
> > > **How do the authors identify where oracle interventions should be pinpointed? How to control the length of each generated intervention?**
> >
> > The locations for oracle interventions are determined automatically by the oracle models: they verify the solution **step-by-step** and halt as soon as an error is detected. The underlying intuition is that **LLMs are often better at verifying solutions than generating them**, allowing the oracle to reliably identify mistakes and produce targeted interventions.
> >
> > We control the length of each intervention through prompting - specifically, by instructing the model to generate only a **single corrective step** (see Appendix A for the prompt). In practice, this works well: the interventions are consistently much shorter than full solutions, as shown in Figure 3.

---

> > > ### Author Response · Authors · 2025-11-26
> > > **Did our Rebuttal Address your Concerns?**
> > >
> > > Dear Reviewer,
> > >
> > > Since there is only 1 week left in the rebuttal period, it would be helpful for us to know if our rebuttal above addressed all your questions and concerns or if any others still remained? We are happy to discuss further and provide more evidence if it is helpful. If all your concerns are addressed we would be grateful if you could acknowledge that soon.
> > >
> > > Thanks,
> > >
> > > Authors

---

> > > > ### Author Response · Authors · 2025-11-27
> > > >
> > > > Dear Reviewer,
> > > >
> > > > We apologize for pinging you again!
> > > >
> > > > Given that there are only a four more days left in the rebuttal period, we wanted to reach out once more to see if our rebuttal above addressed all your concerns or if any others still remained? We are happy to discuss further and provide more evidence in the few days that are left. If all your concerns are addressed we would be really grateful if you could acknowledge that soon.
> > > >
> > > > Thanks,
> > > >
> > > > Authors

---

### Official Review · Reviewer_ufkp · 2025-11-01

**Soundness:** 2
**Presentation:** 3
**Contribution:** 2
**Rating:** 4
**Confidence:** 4

**Summary:**

This work tackles a highly valuable problem, the failure of reinforcement learning (RL) to improve large language models (LLMs) on “hard” reasoning tasks where no correct rollouts are sampled, resulting in zero reward and stalled learning. The authors propose Interventional Training (InT), which introduces targeted external interventions at the first failure point in a model’s reasoning chain. By forcing a corrected trajectory and continuing RL from this modified trace, the method effectively injects learning signals into regions where RL previously could not make progress.

**Strengths:**

1. The proposed method directly tackles the zero-reward problem in RL for reasoning models, enabling continued learning even beyond the model’s existing competence boundary.
2. By identifying and correcting the first erroneous step in the reasoning chain, InT provides much finer-grained credit assignment than standard RL, improving learning efficiency and self-correction capability.

**Weaknesses:**

1. The core idea is not new — applying single-step interventions at failure points has already been explored in other RL and imitation-learning domains. The paper mainly transfers this known concept to LLM reasoning without introducing new techniques.
2. The method depends on access to ground-truth answers and a strong evaluation model to identify and correct errors, raising concerns about scalability and whether such oracle-dependent training has an inherent upper bound on achievable improvement.
3. The evaluation primarily compares InT against ablated variants such as Distillation + RL and Standard RL, rather than against more conceptually related methods like process-reward modeling or critique-based self-refinement, which limits the persuasiveness of the results.

**Questions:**

Please refer to the concerns in Weaknesses.

---

> ### Author Response · Authors · 2025-11-24
> **Rebuttal by the Authors (Part 1/2)**
>
> Thank you for the review!  Since the original submission, we have made substantial updates to the work. Most importantly, we have:
>
> 1. Added several new experiments that **remove entirely the dependence on an external oracle model (Gemini 2.5 Pro).** The base model now generates its own interventions, which clarifies the core contributions of InT: it now becomes clear that InT is not simply distilling responses from a stronger teacher. Instead, InT benefits from external signals (gold solutions) to produce effective interventions, highlighting it as a self-contained training strategy. We will include these new experiments as the main results of the paper.
>
> 2. **Introduced a new critique-based self-refinement baseline as requested,** where the base model revises its entire solution, rather than producing a localized single-step intervention. A full description and the exact prompt used for self-refinement are provided in **Appendix J**.
>
> 3. **Evaluated models on more challenging, standardized benchmarks:** IMO-Bench [1], AMO-Bench [2], and Apex Shortlist [3], and the newly released, uncontaminated HMMT November 2025 problems, which we scraped from [4]. We run RL for *400 steps* and observe that training with InT leads to a near 10% improvement in average performance compared to the base model, and 3% improvement compared to the strongest baseline.
>
> Below is a table summarizing our final results using **Qwen3-4B-Instruct** as the base model:
>
> | Model    | IMO-Bench | HMMT 2025 Nov | AMO-Bench pass@8 | Apex Shortlist pass@8 | Average |
> |------------------------|-----------|---------------|-------------------|------------------------|---------|
> | Base | 11.68 | 41.61 | 26.24 | 20.79 | 25.08 |
> | + RL                | 23.46     | 46.46         | 35.21             | 22.72                  | 31.96   |
> | + Gold SFT + RL          | 11.56     | 27.45         | 25.19             | 20.51                  | 21.18   |
> | + Self-refine SFT + RL   | 15.53     | 38.65         | **36.72**         | 23.93                  | 28.71   |
> | **+ InT + RL (Ours)**     | **25.62** | **49.77**     | 36.16             | **28.22**              | **34.94**   |
>
> We also show the **pass@k curves** across training on the test set **in the figure below** to evaluate if InT is meaningfully able to go beyond the capabilities of the base model or simply “sharpens” what the base model already knows. Across both sets of results, **InT achieves the strongest overall performance and a higher pass@k value for all $k$ values we could evaluate**, indicating that patching the LLM with InT indeed improves its overall capabilities.
>
> **Figure:** [Pass@k curves over training](https://ibb.co/pvWfKqH3)
>
> If these revisions address your concerns, we kindly request that you consider updating your score. Thank you!
>
> [1] [IMO-Bench](https://imobench.github.io/)
>
> [2] [AMO-Bench](https://amo-bench.github.io/)
>
> [3] [Apex Shortlist](https://huggingface.co/datasets/MathArena/apex-shortlist)
>
> [4] [HMMT Website](https://www.hmmt.org/www/archive/problems)

---

> ### Author Response · Authors · 2025-11-24
> **Rebuttal by the Authors (Part 2/2)**
>
> We answer your questions in more detail below.
>
> > **The core idea is not new — applying single-step interventions at failure points has already been explored in other RL and imitation-learning domains. The paper mainly transfers this known concept to LLM reasoning without introducing new techniques.**
>
> Although the high-level idea of single-step interventions is not new in standard human-in-the-loop imitation learning for robot control or self-driving (e.g., HG-DAgger [1], see “Intervention training outside of LLMs.” in Section 2 for more related works), to the best of our knowledge, **our work is the first to apply it to LLM reasoning and to show that this strategy substantially improves performance and patches the gaps in reasoning.** This represents a meaningful contribution not only because we improve results for LLM reasoning but also because of several novel aspects of applying interventions to LLMs: LLMs allow *precisely specified* corrections to the first reasoning error, something we cannot quite do in robotics or self-driving; InT exploits the property unique to LLMs that verifying steps (against the gold solution, to detect the location to insert interventions) is often easier than generating them, enabling models to improve through **self-generated** interventions, which is also absent in robotics or other areas where interventions have been used. **If you think that there is some piece of work that is close to our paper, please point out that work precisely and we will be happy to cite and compare against it.**
>
> [1] HG-DAgger: https://arxiv.org/abs/1810.02890
>
> > **The method depends on access to ground-truth answers and a strong evaluation model to identify and correct errors, raising concerns about scalability and whether such oracle-dependent training has an inherent upper bound on achievable improvement.**
>
> In our updated experiments, we have **completely removed the dependence on external oracles (Gemini 2.5 Pro)**. All interventions are now generated **by the base model itself** (Qwen3-4B-Instruct). This avoids the cost and access limitations of closed-source oracles and makes the method much more practical for anyone to use, as long as they have access to the gold solution to a problem (which often is the case for most reasoning datasets). We’ve included a short summary of the new results above, with more details in Appendix H.
>
> Regarding **gold (ground-truth) answers**, our experiments still assume access to reference solutions for training, which is standard practice in the math reasoning domain. For example, datasets such as Omni-MATH and DeepScaleR consist of a wide variety of problems annotated with human-written “reference” solutions, and **these solutions are all that is needed for our method to work.**  Handling tasks **without** gold answers - where correctness is uncertain or must be inferred - is an important direction, but outside the scope of this work and remains a promising area for future research. We will add this discussion in the paper. That said, even showing that interventions can address gaps in an LLM’s reasoning on problems where answers are available is already quite important and a strong first step, hence we believe this should not be a reason to reject our work.
>
> > **The evaluation primarily compares InT against ablated variants such as Distillation + RL and Standard RL, rather than against more conceptually related methods like process-reward modeling or critique-based self-refinement, which limits the persuasiveness of the results.**
>
> To address this concern, we discuss two closely related methods:
>
> 1. **Self-refinement baseline.** We implemented a self-refinement method where the model critiques its full reasoning chain and revises its answer, making it directly comparable to InT (implementation details in **Appendix J**). As shown the table above (**more details in Appendix H**), InT consistently outperforms self-refinement, perhaps indicating that **targeted first-error interventions provide clearer credit assignment than global critiques**, which may dilute the learning signal.
>
> 2. **PRMs.** PRMs are relevant, but high-quality PRMs are not yet available for the newer base models we use and prior work has shown that PRMs underperform pure RL baselines, especially on difficult problems [1], and training a PRM from scratch requires significant additional work. We can include experiments using an available open-source option (e.g., Qwen2.5-Math-PRM-7B) if the reviewer feels this comparison is important, and would welcome guidance on the setup they consider most appropriate, and we will be happy to add it in the remainder of this rebuttal period.
>
> [1] [​​Is PRM Necessary? Problem-Solving RL Implicitly Induces PRM Capability in LLMs](https://arxiv.org/abs/2505.112)

---

> > ### Author Response · Authors · 2025-11-26
> > **Did our Rebuttal Address your Concerns?**
> >
> > Dear Reviewer,
> >
> > Since there is only 1 week left in the rebuttal period, it would be helpful for us to know if our rebuttal above addressed all your questions and concerns or if any others still remained? We are happy to discuss further and provide more evidence if it is helpful. If all your concerns are addressed we would be grateful if you could acknowledge that soon.
> >
> > Thanks,
> >
> > Authors

---

> > > ### Author Response · Authors · 2025-11-27
> > >
> > > Dear Reviewer,
> > >
> > > We apologize for pinging you again!
> > >
> > > Given that there are only a four more days left in the rebuttal period, we wanted to reach out once more to see if our rebuttal above addressed all your concerns or if any others still remained? We are happy to discuss further and provide more evidence in the few days that are left. If all your concerns are addressed we would be really grateful if you could acknowledge that soon.
> > >
> > > Thanks,
> > >
> > > Authors

---

### Official Review · Reviewer_jykS · 2025-11-02

**Soundness:** 3
**Presentation:** 3
**Contribution:** 3
**Rating:** 4
**Confidence:** 4

**Summary:**

This paper introduces Interventional Training, a novel framework designed to address credit assignment issues in LLM reasoning during reinforcement learning. The key idea is to leverage single-step oracle interventions (e.g., from a human or another LLM) to correct intermediate errors in model-generated reasoning traces, followed by supervised fine-tuning (SFT) and continued RL. The authors demonstrate that InT improves performance on hard problems where standard RL fails, achieving gains in pass@k metrics and solving previously unsolvable tasks. The contributions include a cost-effective alternative to full trace cloning and a method to patch LLMs without distorting their base distributions.

**Strengths:**

InT creatively uses localized oracle interventions to patch errors, avoiding the pitfalls of full trace distillation.
Experimental results show consistent improvements in pass@k and problem-solving rates, with ablations validating design choices.

**Weaknesses:**

The primary weakness is the reliance on an external oracle model (e.g., Gemini 2.5 Pro) for interventions. This introduces practical constraints, such as the cost and availability of high-performance oracles, and potential biases if the oracle's capabilities do not generalize.
While InT reduces data-writing burden compared to full traces, it still requires oracle access, which may not be feasible for all practitioners.

**Questions:**

How can the dependence on an external oracle be minimized? For instance, could the base model be trained to self-correct interventions over time?
What are the trade-offs in using different oracle types (e.g., humans vs. LLMs), and how do they impact reproducibility?

---

> ### Author Response · Authors · 2025-11-24
> **Rebuttal by the Authors (Part 1/2)**
>
> Thank you for the review! We have updated our work significantly since the submission. Most notably, we have
>
> 1. Added several new experiments that **remove entirely the dependence on an external oracle model (Gemini 2.5 Pro).** The base model now generates its own interventions, which clarifies the core contributions of InT: it now becomes clear that InT is not simply distilling responses from a stronger teacher. Instead, InT benefits from external signals (gold solutions) to produce effective interventions, highlighting it as a self-contained training strategy. We will include these new experiments as the main results of the paper.
>
> 2. **Evaluated models on more challenging, standardized benchmarks:** IMO-Bench [1], AMO-Bench [2], and Apex Shortlist [3], and the newly released, uncontaminated HMMT November 2025 problems, which we scraped from [4]. We run RL for *400 steps* and observe that training with InT leads to a near 10% improvement in average performance compared to the base model, and 3% improvement compared to the strongest baseline.
>
> Below is a table with our final results using **Qwen3-4B-Instruct** as the base model:
>
> | Model    | IMO-Bench | HMMT 2025 Nov | AMO-Bench pass@8 | Apex Shortlist pass@8 | Average |
> |------------------------|-----------|---------------|-------------------|------------------------|---------|
> | Base | 11.68 | 41.61 | 26.24 | 20.79 | 25.08 |
> | + RL                | 23.46     | 46.46         | 35.21             | 22.72                  | 31.96   |
> | + Gold SFT + RL          | 11.56     | 27.45         | 25.19             | 20.51                  | 21.18   |
> | + Self-refine SFT + RL   | 15.53     | 38.65         | **36.72**         | 23.93                  | 28.71   |
> | **+ InT + RL (Ours)**     | **25.62** | **49.77**     | 36.16             | **28.22**              | **34.94**   |
>
> We also show the **pass@k curves** across training on the test set **in the figure below** to evaluate if InT is meaningfully able to go beyond the capabilities of the base model or simply “sharpens” what the base model already knows. Across both sets of results, **InT achieves the strongest overall performance and a higher pass@k value for all $k$ values we could evaluate**, indicating that patching the LLM with InT indeed improves its overall capabilities.
>
> **Figure:** [Pass@k curves over training](https://ibb.co/pvWfKqH3)
>
> Note: in the results above, **self-refinement** refers to a new baseline where the base model revises its entire solution rather than producing a localized single-step intervention. This baseline is important because it isolates the value of *targeted* interventions: if simple whole-solution rewriting were sufficient, self-refinement should match or exceed InT. Instead, our results show that localized interventions provide a much stronger training signal by focusing the model on the specific point of failure in its reasoning. A full description and the exact prompt used for self-refinement are provided in **Appendix J**.
>
> If these updates address your concerns, we would be very grateful if you could raise your score. Thank you!
>
> [1] [IMO-Bench](https://imobench.github.io/)
>
> [2] [AMO-Bench](https://amo-bench.github.io/)
>
> [3] [Apex Shortlist](https://huggingface.co/datasets/MathArena/apex-shortlist)
>
> [4] [HMMT Website](https://www.hmmt.org/www/archive/problems)

---

> ### Author Response · Authors · 2025-11-24
> **Rebuttal by the Authors (Part 2/2)**
>
> We answer your questions in more detail below.
>
> > **The primary weakness is the reliance on an external oracle model (e.g., Gemini 2.5 Pro) for interventions. This introduces practical constraints, such as the cost and availability of high-performance oracles, and potential biases if the oracle's capabilities do not generalize. While InT reduces data-writing burden compared to full traces, it still requires oracle access, which may not be feasible for all practitioners.**
>
> > **How can the dependence on an external oracle be minimized? For instance, could the base model be trained to self-correct interventions over time?**
>
> We have addressed this concern in our updated experiments which generates interventions **without any dependence on external oracles (Gemini 2.5 Pro)**. Instead, the **base model now generates its own interventions**, relying only on an openly available, open-source model. This removes the cost and availability constraints of proprietary oracles and opens up the method to practitioners who do not have access to them.
>
> A brief summary of our main results is provided above, with further details in **Appendix H**. Taken together with the results already present in the paper, these new results highlight that InT does not need to rely on oracles for interventions. Perhaps because verifying an existing solution against a gold solution is easier than generating a solution from scratch, we find that using base models as the oracle is sufficient for generating interventions.
>
> > **What are the trade-offs in using different oracle types (e.g., humans vs. LLMs), and how do they impact reproducibility?**
>
> In principle, one can argue that humans training frontier models do provide oracle information in several forms such as curating training prompts, designing the right reward functions, etc. In the case of InT, relying on humans to **directly generate interventions** presents both advantages and drawbacks. Although a human annotator would still need to read long reasoning chains to locate the correct point of intervention, providing short interventions may still be easier than producing complete solutions from scratch.
>
> That said, our new results show that we can *sidestep* the requirement of reading a long reasoning trace by **using the base LLM to self-generate interventions conditioned on gold solutions provided by a human.** These gold solutions need not be tailored to the model at hand, nor do they need to be in the format of a reasoning trace, and are readily available in many existing math datasets (e.g., DeepScaleR [1], Omni-MATH [2], etc.). InT then relies on the property of LLMs that verification (of an existing solution against a gold solution) is easier than generation (of a solution from scratch) to self-generate effective interventions.
>
> **Figure:** [InT facilitates on-policy learning](https://ibb.co/KcYqpRQ4)
>
> A side benefit of this InT is that it relies on **on-policy learning**, unlike SFT on gold solutions alone. On-policy training is known to improve compute efficiency and reduce forgetting [3, 4] or other pathologies with imitating data from proprietary models [5]. InT sidesteps these concerns since learning mostly happens on tokens generated by the base model itself. **As shown in the figure above**, InT traces have the highest likelihood under the base model and training with InT leads to the best train and test pass@k accuracies across all k measured. Additional details on this analysis are in **Appendix I**.
>
> [1] [DeepScaleR blog](https://pretty-radio-b75.notion.site/DeepScaleR-Surpassing-O1-Preview-with-a-1-5B-Model-by-Scaling-RL-19681902c1468005bed8ca303013a4e2)
>
> [2] [Omni-MATH](https://omni-math.github.io/)
>
> [3] [Thinking Machines on-policy distillation blog](https://thinkingmachines.ai/blog/on-policy-distillation/)
>
> [4] [Retaining by Doing: The Role of On-Policy Data in Mitigating Forgetting](https://arxiv.org/abs/2510.18874)
>
> [5] [The False Promise of Imitating Proprietary LLMs](https://arxiv.org/abs/2305.15717)

---

> > ### Author Response · Authors · 2025-11-26
> > **Did our Rebuttal Address your Concerns?**
> >
> > Dear Reviewer,
> >
> > Since there is only 1 week left in the rebuttal period, it would be helpful for us to know if our rebuttal above addressed all your questions and concerns or if any others still remained? We are happy to discuss further and provide more evidence if it is helpful. If all your concerns are addressed we would be grateful if you could acknowledge that soon.
> >
> > Thanks,
> >
> > Authors

---

> > > ### Comment · Reviewer_jykS · 2025-11-27
> > >
> > > The authors have resolve most of my concerns, and I have raise the score.

---

### Author Response · Authors · 2025-12-02
**Summary of Rebuttal Updates for Area Chair**

Dear Area Chair,

Thank you for overseeing the review process. Given the reviewer-data issue and the updated policy that decisions will now rely on pre-rebuttal scores, we wanted to provide a clear summary of the reviewers' main concerns and how we addressed them during the rebuttal period.

**Summary of reviewer follow-ups and score changes**

Of the four reviewers, three (with initial scores of 4, 4, and 6) have not had a chance to follow up yet. One reviewer did respond that our rebuttal resolved their concerns and subsequently raised their score from 4 to 6.

We would like to point out that our rebuttal directly addressed a *major concern that was shared by all reviewers*, and this led to a significant improvement of the proposed method itself. Because of this, we believe that, had the rebuttal period run its full course, there is a strong likelihood they would also have raised their scores.

**Main concerns of the reviewers**
> This work requires access to a stronger oracle model (i.e., Gemini 2.5 Pro).

This has been the *most dominant* concern raised by all reviewers.

We have fully addressed this concern in our new experiments by *removing entirely the dependence on an external oracle model (Gemini 2.5 Pro). Instead, the base model now generates its own interventions,* which clarifies the core contributions of InT: it now becomes clear that InT is not simply distilling responses from a stronger teacher. Instead, InT benefits from external signals (gold solutions) to produce effective interventions, highlighting it as a self-contained training strategy. These new experiments will now be included as the *main results* of the paper.

> Baselines considered are not strong enough.

As many of the reviewers have requested for a critique-based self-refinement baseline, we have added it to our rebuttal. In self-refinement, the base model revises its entire solution rather than producing a localized single-step intervention. A full description and the exact prompt used for self-refinement are provided in Appendix J.

> Experimental validations are insufficient.

To this end, we evaluated our new approach with *self-generated interventions* on more challenging, standardized benchmarks: IMO-Bench [1], AMO-Bench [2], and Apex Shortlist [3], and the newly released, uncontaminated HMMT November 2025 problems, which we scraped from [4]. We run RL for 400 steps and observe that training with InT leads to a near 10% improvement in average performance compared to the base model, and 3% improvement compared to the strongest baseline.

Below is a table with our final results using **Qwen3-4B-Instruct** as the base model:

| Model    | IMO-Bench | HMMT 2025 Nov | AMO-Bench pass@8 | Apex Shortlist pass@8 | Average |
|------------------------|-----------|---------------|-------------------|------------------------|---------|
| Base | 11.68 | 41.61 | 26.24 | 20.79 | 25.08 |
| + RL                | 23.46     | 46.46         | 35.21             | 22.72                  | 31.96   |
| + Gold SFT + RL          | 11.56     | 27.45         | 25.19             | 20.51                  | 21.18   |
| + Self-refine SFT + RL   | 15.53     | 38.65         | **36.72**         | 23.93                  | 28.71   |
| **+ InT + RL (Ours)**     | **25.62** | **49.77**     | 36.16             | **28.22**              | **34.94**   |

We also show the pass@k curves across training on the test set in the figure below to evaluate if InT is meaningfully able to go beyond the capabilities of the base model or simply “sharpens” what the base model already knows. Across both sets of results, *InT achieves the strongest overall performance and a higher pass@k* for all $k$ values we could evaluate, indicating that patching the LLM with InT indeed improves its overall capabilities.

**Figure:** [Pass@k curves over training](https://ibb.co/pvWfKqH3)

We hope this summary is helpful, and please let us know if any further clarification would assist in your decision-making. Thank you!

[1] [IMO-Bench](https://imobench.github.io/)

[2] [AMO-Bench](https://amo-bench.github.io/)

[3] [Apex Shortlist](https://huggingface.co/datasets/MathArena/apex-shortlist)

[4] [HMMT Website](https://www.hmmt.org/www/archive/problems)

---

### Meta-Review · Area_Chair_dF6M · 2025-12-24

**Summary:**

This paper introduces Interventional Training (InT), a framework designed to overcome the zero-reward limitation in reinforcement learning for large language models. By leveraging single-step oracle interventions to identify and correct intermediate reasoning errors, the method performs precise credit assignment on hard problems. The model is patched using supervised fine-tuning on these corrected traces before resuming reinforcement learning. Experimental results demonstrate that InT significantly improves problem-solving rates and performance across various mathematical reasoning benchmarks.

**Strengths:**
1. The method effectively addresses the critical zero-reward bottleneck in reinforcement learning by enabling models to make progress on problems beyond their current competence boundaries. It provides a more fine-grained approach to credit assignment than standard reinforcement learning by targeting the first erroneous step in a reasoning chain.
2. InT is a lightweight and cost-effective  because it requires only a small number of short oracle interventions. By only patching on-policy prefixes, the approach avoids catastrophic forgetting and maintains model diversity better than traditional imitation learning methods.
3. The paper is well-organized and clearly demonstrates consistent performance gains across multiple math reasoning benchmarks and hard test problems. The authors provide thorough ablations and analysis of training dynamics that validate their core design choices and the effectiveness of the patching mechanism.

**Weaknesses:**
1. The framework relies heavily on access to a high-performance external oracle for identifying and correcting errors, which may limit its scalability and practical availability. There is also a concern regarding whether this dependency creates an inherent upper bound on performance based on the oracle's own capabilities. The paper needs to adequately explore how the method performs with weaker or human oracles and whether the model truly learns to self-correct rather than just memorizing interventions.
2. The experimental evaluation is somewhat narrow as it focuses primarily on math reasoning tasks and lacks comparisons with conceptually related methods like process-reward modeling. Furthermore, the limited number of hard problems used in the reinforcement learning phase raises questions about the stability and generalizability of the results.
3. Literature review is insufficient, with inadequate discussion of prior work on oracle guidance.

**Reviewer Concerns:**

The primary concern raised by all reviewers was the work's dependence on a strong external oracle model. To address this, the authors have eliminated the reliance on Gemini 2.5 Pro in their updated experiments. Specifically, they introduced a critique-based self-refinement baseline and evaluated the proposed approach using self-generated interventions on more rigorous, standardized benchmarks, including IMO-Bench, AMO-Bench, and Apex Shortlist.

**Reviewer Scores:**

Regarding the review process, Reviewer jykS acknowledged that most concerns have been resolved and expressed a willingness to increase the score. While the other three reviewers (with initial scores of 4, 4, and 6) have not yet followed up, the authors' comprehensive revisions suggest a strong potential for these scores to be improved as well.

---

### Decision · Program_Chairs · 2026-01-26

Accept (Poster)